# Searching for dark matter with a spin-based interferometer

**Daniel Gavilan-Martin** [1,2,3,13] ✉, **Grzegorz Łukasiewicz** [4,5,13] ✉,
**Mikhail Padniuk** [4], **Emmanuel Klinger** [6], **Magdalena Smolis** [4],
**Nataniel L. Figueroa** [1,2], **Derek F. Jackson Kimball** [7],
**Alexander O. Sushkov** [8,9,10,11], **Szymon Pustelny** [4], **Dmitry Budker** [1,2,3,12] &
**Arne Wickenbrock** [1,2,3]

Axion-like particles (ALPs) arise from well-motivated extensions to the Standard Model and could account for dark matter. ALP dark matter would manifest as a field oscillating at an (as of yet) unknown frequency. The frequency depends linearly on the ALP mass and plausibly ranges from $10^{-22}$ to $10$ eV/$c^2$. This motivates broadband search approaches. We report on a direct search for ALP dark matter with an interferometer composed of two atomic K-Rb-$^3$He comagnetometers, one situated in Mainz, Germany, and the other in Kraków, Poland. We leverage the anticipated spatio-temporal coherence properties of the ALP field and probe all ALP-gradient-spin interactions covering a mass range of nine orders of magnitude. No significant evidence of an ALP signal is found. We thus place new upper limits on the ALP-neutron, ALP-proton and ALP-electron couplings reaching below $g_{aNN} < 10^{-9}$ GeV$^{-1}$, $g_{aPP} < 10^{-7}$ GeV$^{-1}$ and $g_{aee} < 10^{-6}$ GeV$^{-1}$, respectively. These limits improve upon previous laboratory constraints for neutron and proton couplings by up to three orders of magnitude.

Abundant astrophysical and cosmological observations at different scales[1–3] suggest that about 85% of the matter in the Universe does not noticeably interact beyond gravitational interactions and is thus known as "dark" matter (DM). Since the DM composition remains unknown, it serves as a provocative indication of physics beyond the Standard Model (SM) and drives the search for hypothetical DM particles.

Ultralight ($\lesssim 10$ eV/$c^2$) pseudoscalar bosons known as axion-like particles (ALPs) are particularly well-motivated DM candidates[4–7]. ALPs could account for the correct abundance of relic DM (via, for example,

a misalignment mechanism[8]), and a subset of them could resolve the strong-CP problem[9]. Additionally, ALPs can interact with SM particles through couplings to photons, gluons, and fermions[10], offering direct ways of probing their existence[11–13].

In this article, we present the results of an interferometric ALP search (see also refs. [14,15]). The search explores a wide mass range (nine orders of magnitude) and investigates ALP couplings to three distinct spin types (i.e., those associated with proton, neutron, and electron). The interferometer is composed of two K-Rb-$^3$He atomic comagnetometers, one located at the Jagiellonian University in

¹Johannes Gutenberg-Universität Mainz, Mainz, Germany. ²Helmholtz Institute Mainz, Mainz, Germany. ³GSI Helmholtzzentrum für Schwerionenforschung GmbH, Darmstadt, Germany. ⁴Marian Smoluchowski Institute of Physics, Jagiellonian University in Kraków, Kraków, Poland. ⁵Doctoral School of Exact and Natural Sciences, Jagiellonian University in Kraków, Kraków, Poland. ⁶Institut FEMTO-ST—UMR 6174 CNRS, SupMicroTech-ENSMM, Université de Franche-Comté, Besançon, France. ⁷Department of Physics, California State University—East Bay, Hayward, CA, USA. ⁸Department of Physics, Boston University, Boston, MA, USA. ⁹Department of Electrical and Computer Engineering, Boston University, Boston, MA, USA. ¹⁰Photonics Center, Boston University, Boston, MA, USA. ¹¹Department of Physics & Astronomy, The Johns Hopkins University, Baltimore, MD, USA. ¹²Department of Physics, University of California, Berkeley, CA, USA. ¹³These authors contributed equally: Daniel Gavilan-Martin, Grzegorz Łukasiewicz. ✉e-mail: gaviland@uni-mainz.de; grzegorz.lukasiewicz@doctoral.uj.edu.pl

Kraków, Poland[16] and the other, separated by 860 km, at the Johannes Gutenberg University in Mainz, Germany, see Fig. 1a. The interference occurs through the phase-sensitive combination of the amplitude data, in a manner similar to that used in radio telescope networks (e.g., the event horizon telescope[17]). The main difference between our and other schemes is that we are sensitive to ALP DM gradients rather than electromagnetic waves, and that the corresponding improvement in angular resolution does not play a role in searching for DM, which is assumed to have a homogeneous local density. It might, however, be important in searches when DM models feature more heterogeneous mass distribution, including streams and halos[18–21], or if there are distinguishable emitters of bosonic fields in the Universe, e.g., black holes emitting ALPs in a process of superradiance[22] or during binary merger[23]. The interference increases the signal-to-noise ratio of the search compared to a single sensor[24]. This work constitutes the first constraints from an interferometer composed of two comagnetometers.

We optimize both comagnetometers to operate in the self-compensating regime[25–27]. To a first order, the devices are insensitive to low-frequency magnetic-field variations, but retain sensitivity to non-magnetic spin interactions[28,29]. This makes them excellent tools for probing the interaction of the galactic ALP DM field $a(r, t)$ with neutron spins $\sigma_N$, proton spins $\sigma_P$ and electron spins $\sigma_e$, which are described by the Hamiltonians

$$
\begin{aligned}
\mathcal{H}_N &= g_{aNN} \nabla a \cdot \sigma_N , \\
\mathcal{H}_P &= g_{aPP} \nabla a \cdot \sigma_P , \\
\mathcal{H}_e &= g_{aee} \nabla a \cdot \sigma_e ,
\end{aligned}
\tag{1}
$$

where $g_{aNN}$, $g_{aPP}$, and $g_{aee}$ are unknown coupling constants to neutrons, protons, and electrons. Self-compensating comagnetometers have

already established the most stringent limits in certain mass ranges, even surpassing astrophysical constraints[30–32]. In this work, we search in experimentally unconstrained ALP parameter space in the ultralight mass range below $10^{-13}$ eV/$c^2$.

If the estimated local DM density ($\approx 0.4$ GeV/cm$^3$)[33] is mostly due to an ALP of mass $m_a$, the occupation number of the ALP field would be large, and hence it can be approximated as a classical field oscillating near the ALP Compton frequency[34]. In this model, the characteristics of the oscillating ALP field, such as the amplitudes and the phases of the ALP-gradient components, fluctuate in time according to the properties of virialized DM[35–39]. The characteristic time scale of the fluctuations results from the ALP DM velocity spread, which leads to a coherence time of around $10^6$ oscillations of the field (for example, about 15 years for $m_a = 10^{-17}$ eV/$c^2$).

In our work, we focus primarily on the regime where the coherence time of the ALP collective oscillation is larger than the total measurement time. In this regime, the ALP field can be treated as having a constant amplitude and direction, and hence its signatures at the two sensor locations are highly correlated. For reference, an ALP particle of mass $10^{-17}$ eV/$c^2$, assuming a relative velocity equal to that of Earth in the galactic rest frame, has a de Broglie wavelength of $\sim 10^3$ astronomical units. Thus, by properly combining the signals from both stations, the ALP signals will be added constructively, while the effects of local noise fluctuations sum incoherently and are suppressed, see Fig. 1c.

An additional analysis of the results allows us to extend the search to ALP masses below $\sim 5 \times 10^{-20}$ eV/$c^2$. This is possible by looking for spectral signatures in the data that would arise due to the rotation of the Earth. The properties of the field gradient in this ultra-low frequency regime are discussed, and the signal properties are explicitly reviewed below.

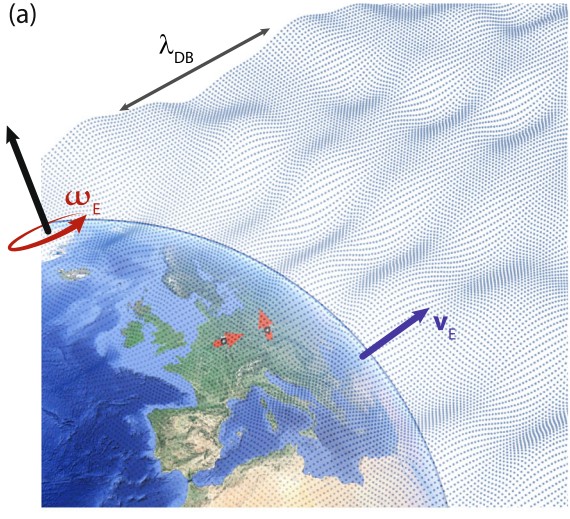

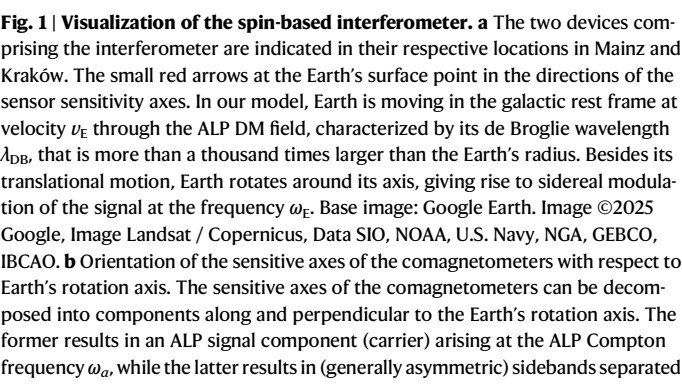

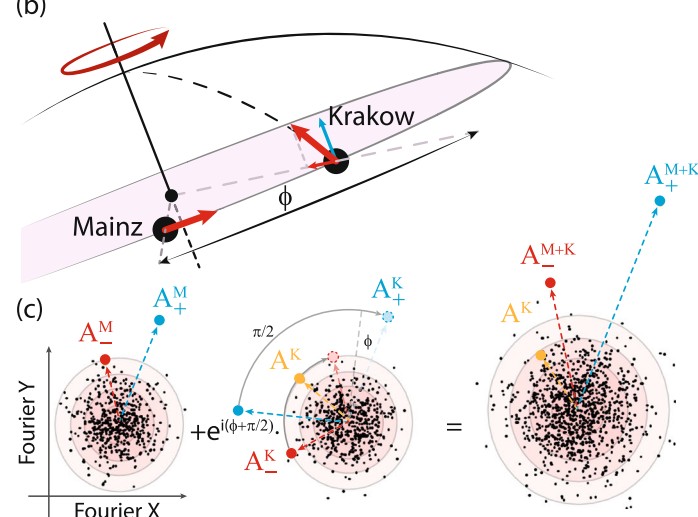

**Fig. 1 | Visualization of the spin-based interferometer. a** The two devices comprising the interferometer are indicated in their respective locations in Mainz and Kraków. The small red arrows at the Earth's surface point in the directions of the sensor sensitivity axes. In our model, Earth is moving in the galactic rest frame at velocity $v_E$ through the ALP DM field, characterized by its de Broglie wavelength $\lambda_{DB}$, that is more than a thousand times larger than the Earth's radius. Besides its translational motion, Earth rotates around its axis, giving rise to sidereal modulation of the signal at the frequency $\omega_E$. Base image: Google Earth. Image ©2025 Google, Image Landsat / Copernicus, Data SIO, NOAA, U.S. Navy, NGA, GEBCO, IBCAO. **b** Orientation of the sensitive axes of the comagnetometers with respect to Earth's rotation axis. The sensitive axes of the comagnetometers can be decomposed into components along and perpendicular to the Earth's rotation axis. The former results in an ALP signal component (carrier) arising at the ALP Compton frequency $\omega_a$, while the latter results in (generally asymmetric) sidebands separated from the carrier by the sidereal frequency $\omega_E$. **c** Signal interferometry in the data analysis (illustration). The points in the three subfigures correspond to the complex Fourier amplitudes of all probed frequency bins of the Mainz (left), Kraków (middle) datasets, and their combination (right). The frequency points are intended for illustration purposes and do not correspond to experimental data. We assume normal noise distributions. The circles indicate the standard deviation. The points marked with red, blue, and orange represent injected ALP signatures observable in both comagnetometers and how they appear in the combined signal. Due to the directional sensitivity of the comagnetometers, the injected ALP signal manifests as a carrier of amplitude $A^K$ only in the Kraków data, and sideband signatures of different amplitudes and phases in both Kraków ($A^K_\pm$) and Mainz ($A^M_\pm$) data. The phase difference between the signals arises due to the different orientation of the sensitive axes ($\pi/2$), as well as the different locations ($\phi$) of the sensors. Appropriate phasing allows to coherently add the ALP signals, while the noise adds incoherently.

This article is structured as follows. In the section "Signal model," we review the ALP DM signal model, explicitly showing the expected signatures in the frequency domain for a daily modulated sensor, and discuss how the sensitivity can be improved by analyzing interfering signals from multiple sensors. In the section "Search strategy and results," we show the analysis framework and discuss how data is combined to maximize a signal-to-noise ratio. As no ALP candidates are found, we set limits on the ALP DM pseudoscalar spin interactions in the section "Setting limits." The experimental setup and some technical details of the search are presented in the section "Methods."

## Results and discussion

### Signal model

To describe the expected signal model in the two comagnetometer locations, we utilize the framework described in refs. [30,37,38]. Due to its high occupation number, the ALP field can be approximated as a sum of $N$ independent oscillations. We write the components of the ALP field gradient at position $\mathbf{r}$ and time $t$ in a Cartesian frame of reference $i = x, y, z$, where the Solar System is at rest

$$
\begin{aligned}
\boldsymbol{\nabla}_i a(\mathbf{r}, t) &= \frac{\hbar \sqrt{2\rho_{\mathrm{DM}}}}{m_a c \sqrt{N}} \sum_{n=1}^{N} \boldsymbol{\nabla}_i \sin(\omega_n t - \mathbf{k}_n \cdot \mathbf{r} + \phi'_n) \\
&= \frac{\sqrt{2\rho_{\mathrm{DM}}}}{c \sqrt{N}} \sum_{n=1}^{N} (\mathbf{v}_n)_i \cos(\omega_n t - \mathbf{k}_n \cdot \mathbf{r} + \phi_n).
\end{aligned}
\tag{2}
$$

The negative sign in the second line is absorbed by the random phase $\phi_n = \phi'_n + \pi$, $c$ is the speed of light, $m_a$ is the ALP mass, $\rho_{\mathrm{DM}} \approx 0.4 \, \mathrm{GeV/cm^3}$[33] is the local DM density, $\mathbf{k}_n = m_a \mathbf{v}_n / \hbar$ is the ALP-field wave vector, $\omega_n$ is the angular frequency of each oscillating mode $n$, and $(\mathbf{v}_n)_i$ is the $i$-th component of the relative velocity $\mathbf{v}_n$ between the ALP mode and the sensor. Assuming that ALPs are virialized in the Milky Way, the velocities $\mathbf{v}_n$ follow a 3D normal distribution centered at zero in the galactic frame. On Earth, the observed velocity distribution will be offset by Earth's velocity in the galactic frame $(\mathbf{v}_{\mathrm{E}})_i$. We neglect effects due to the movement of Earth within the Solar System, as it is negligible compared to the velocity of the Solar System in the galaxy[38]. The velocity components $(\mathbf{v}_n)_i$ are then distributed according to

$$
f(v_i) = \frac{1}{v_0 \sqrt{\pi}} \exp\left\{ -\left[\frac{v_i - (\mathbf{v}_{\mathrm{E}})_i}{v_0}\right]^2 \right\},
\tag{3}
$$

where $v_0 \approx 220 \, \mathrm{km/s}$ is the virial velocity in the Milky Way, determining the variance of the velocity distribution to be $v_0^2/2$. The DM escape velocity is neglected, since it has a negligible effect on the distribution.

Each mode of the ALP field experiences a kinetic energy correction, leading to a small frequency shift $\omega_n \approx \omega_a(1 + v_n^2/2c^2)$. Thus, most of the ALP spectrum is concentrated within a spectral width given by[38]

$$
\Delta\omega \approx \omega_a \frac{v_0^2}{c^2} \approx \omega_a \times 10^{-6}.
\tag{4}
$$

The collective mode can be described by a nearly monochromatic oscillation with independent amplitude and phase for each orthogonal spatial direction. The amplitudes and phases are random and evolve smoothly on time scales set by the coherence time $\tau \approx 2\pi/\Delta\omega$. The underlying probability distributions of the amplitudes and phases can be derived from Eqs. (2) and (3)[38]. This results in six independent parameters: three amplitudes ($\alpha_x$, $\alpha_y$, and $\alpha_z$), following Rayleigh distributions, and three phases ($\phi_x$, $\phi_y$, and $\phi_z$), following uniform distributions over $[0, 2\pi)$[37]. Because the ALP de Broglie wavelength in the analyzed mass range is much larger than the sensor separation, $\lambda = 2\pi/k_n \gg d = 860 \, \mathrm{km}$, we neglect the spatial dependence, i.e.,

$\mathbf{k}_n \cdot \mathbf{r} \approx 0$, in Eq. (2) for both sensors. The ALP gradient can then be written as

$$
\begin{aligned}
\boldsymbol{\nabla} a(t) &= \frac{\sqrt{2\rho_{\mathrm{DM}}}}{c \sqrt{N}} \sum_{n=1}^{N} \mathbf{v}_n \cos(\omega_a t + \phi_n) \\
&= \hat{\mathbf{x}} \alpha_x \cos(\omega_a t + \phi_x) + \hat{\mathbf{y}} \alpha_y \cos(\omega_a t + \phi_y) + \hat{\mathbf{z}} \alpha_z \cos(\omega_a t + \phi_z),
\end{aligned}
\tag{5}
$$

where $\omega_a$ is the Compton angular frequency of the ALP. The sum can be evaluated with the central limit theorem, and results in the amplitude terms $\alpha_i$. The probability distributions of the amplitudes $\alpha_i$ are given by[38]

$$
\alpha_i \sim \frac{\sqrt{2\rho_{\mathrm{DM}}}}{c} \sqrt{\frac{v_0^2/2 + (\mathbf{v}_{\mathrm{E}})_i^2}{2}} \, \alpha.
\tag{6}
$$

where $\alpha$ is a Rayleigh distributed random number with scale parameter equal to one.

The sensitive axis $\hat{\mathbf{m}}$ of a single sensor located at Earth's surface rotates with the sidereal frequency of Earth, $\omega_{\mathrm{E}}$. The coordinate system is chosen such that the $\hat{\mathbf{z}}$ component is parallel to the Earth's rotation axis and the $\hat{\mathbf{x}}$ component is perpendicular to the Greenwich meridian. The coordinate system can be assumed static in the galactic rest frame over the timespan of the experiment. This results in a fixed $m_z$ component and daily modulated $m_x$ and $m_y$ components:

$$
\begin{aligned}
\hat{\mathbf{m}}(t) = &\; \hat{\mathbf{x}} \sin\theta \sin(\omega_{\mathrm{E}} t + \phi_{\mathrm{E}}) \\
&+ \hat{\mathbf{y}} \sin\theta \cos(\omega_{\mathrm{E}} t + \phi_{\mathrm{E}}) \\
&+ \hat{\mathbf{z}} \cos\theta,
\end{aligned}
\tag{7}
$$

where $\theta = \angle(\hat{\mathbf{z}}, \hat{\mathbf{m}})$ is the polar angle and $\phi_{\mathrm{E}} = \angle(\hat{\mathbf{x}}, \hat{\mathbf{m}})$ is the azimuthal angle in a spherical Earth coordinate system. The experimental signal is proportional to the projection of the gradient of the ALP field on the sensitive axis of the sensor.

$$
\begin{aligned}
\boldsymbol{\nabla} a(t) \cdot \hat{\mathbf{m}}(t) = &\; \frac{\alpha_x \sin\theta}{2} \left\{ \sin\left[(\omega_a + \omega_{\mathrm{E}})t + \phi_x + \phi_{\mathrm{E}}\right] - \sin\left[(\omega_a - \omega_{\mathrm{E}})t + \phi_x - \phi_{\mathrm{E}}\right] \right\} \\
&+ \frac{\alpha_y \sin\theta}{2} \left\{ \cos\left[(\omega_a + \omega_{\mathrm{E}})t + \phi_y + \phi_{\mathrm{E}}\right] + \cos\left[(\omega_a - \omega_{\mathrm{E}})t + \phi_y - \phi_{\mathrm{E}}\right] \right\} \\
&+ \alpha_z \cos\theta \cos(\omega_a t + \phi_z).
\end{aligned}
\tag{8}
$$

The pseudomagnetic field in the experiment is given by $B_a = \frac{g_{\mathrm{eff}}}{\mu_n} \boldsymbol{\nabla} a \cdot \hat{\mathbf{m}}$, with $\mu_n/2\pi = -32.4 \, \mathrm{MHz/T}$ the gyromagnetic ratio of $^3\mathrm{He}$, and $g_{\mathrm{eff}} = g_{aNN} \xi_N (g_{aPP} \xi_P)$ the effective nucleon coupling taking into account the neutron (proton) content $\xi_N (\xi_P)$[40]. In the case of an electron interaction, the effective coupling is given by the ratio of the gyromagnetic ratios of electron and neutron $g_{\mathrm{eff}} = g_{aee} \gamma_n / \gamma_e$ (see the section "Comagnetometer response to exotic electron interaction").

To obtain the frequency-domain signature of Eq. (8), we describe the three frequency components of the amplitude-modulated signal as a carrier, oscillating at the ALP Compton frequency $\omega_a$ and two sidebands, separated from the carrier by Earth's sidereal frequency $\pm\omega_{\mathrm{E}}$. The magnitudes of the three components (in magnetic-field units) are given by

$$
\begin{aligned}
|A| &= \frac{g_{\mathrm{eff}}}{\mu_n} \alpha_z \cos\theta, \\
|A_-| &= \frac{g_{\mathrm{eff}}}{\mu_n} \frac{\sin\theta}{2} \sqrt{\alpha_x^2 + \alpha_y^2 - 2\alpha_x \alpha_y \sin(\phi_x - \phi_y)}, \\
|A_+| &= \frac{g_{\mathrm{eff}}}{\mu_n} \frac{\sin\theta}{2} \sqrt{\alpha_x^2 + \alpha_y^2 + 2\alpha_x \alpha_y \sin(\phi_x - \phi_y)}.
\end{aligned}
\tag{9}
$$

Because there are three complex amplitudes $A$ and $A_\pm$ that depend differently on six random variables $\alpha_i$ and $\phi_i$, the amplitudes are

independent of one another. It is worth noting that even though the signal in the different frequency components is a result of an amplitude modulation, the sideband amplitudes are generally asymmetric. However, for measurements that average a large number of coherence times, the sideband magnitudes converge to the same value.

Note that the total power of the gradient of the ALP DM signal within a single coherence patch distributed among three frequencies $\omega_a$, $\omega_+$, and $\omega_-$ is

$$\frac{|A|^2}{\cos^2\theta} + \frac{2(|A_+|^2 + |A_-|^2)}{\sin^2\theta} = g_{\text{eff}}^2\left(\alpha_z^2 + \alpha_x^2 + \alpha_y^2\right) = g_{\text{eff}}^2|\boldsymbol{\alpha}|^2 , \quad (10)$$

where $\boldsymbol{\alpha} = (\alpha_x, \alpha_y, \alpha_z)$ gives the 3D amplitude of the ALP DM field.

In our experimental search for ALP DM, it is essential to consider the role of noise in the comagnetometer data. Correlated measurements of ALP DM with two sensors instead of a single sensor lead to a substantial improvement in sensitivity. In fact, the improvement is greater than a factor of $\sqrt{2}$ expected from two measurements of a quantity with uncorrelated noise[24,41]. The reason lies in the respective orientation of the sensitive axes of the sensors. In our configuration, the combination of both sensors allows access to all spatial components of the ALP-gradient signal.

The individual ALP-gradient components are independent random values described by Rayleigh distributions. Probing all of them simultaneously using the ALP DM interferometer configuration enables us to increase the combined signal amplitude and extend the range of the search into lower values of $g_{\text{eff}}$.

The spatial configuration of the two-station ALP DM interferometer is as follows: the sensitive axis of the Mainz station is horizontal in the laboratory pointing East in the Earth rotation plane ($\theta_M = 90° \pm 1°$), and therefore, is exclusively sensitive to the sideband signal $A_\pm$. In contrast, the sensitive axis of the Kraków station is horizontal in the laboratory pointing North ($\theta_K = 50° \pm 1°$). The Kraków comagnetometer is therefore sensitive to all ALP spectral signatures (carrier and sidebands). The uncertainties on the angle are negligible compared to the statistical noise of the experiment.

## Search strategy and results

In our experiment, we searched for ALP DM with Compton frequencies up to 11.6 Hz. In this frequency range, the linewidth of the ALP signal is below the frequency resolution of the analysis method. Additionally, the sensitive axes of the comagnetometers are modulated at the Earth's sidereal frequency, $\omega_E/2\pi \approx 11.6\ \mu$Hz, resulting in resolvable sidebands, see Eq. (9). The expected amplitude ratio between the central peak (carrier) and sidebands is given by the components of the sensitive axes parallel (carrier) and perpendicular (sidebands) to the rotation axis of Earth.

The data analyzed in this work correspond to a total of 40 25-h segments in Mainz and 28 25-h segments in Kraków, collected over a 92-day period between January 6 and April 7, 2024. Data collection was performed as consistently as possible, but several technical interruptions occurred during the measurement campaign. From that data, 25-h continuous data segments were used for the analysis, ensuring sidereal frequency resolution ($\Delta\omega \lesssim \omega_E$).

In general, when searching for unknown dynamic observables, it is crucial to precisely know the sensor frequency response. However, in the case of ALP DM, the frequency response cannot be directly measured. To address this issue, the frequency responses of both comagnetometers to exotic interactions were inferred using the method reported in ref. 29. The method involves (1) measuring the response to a magnetic step perturbation, (2) fitting the Discrete Fourier Transform (DFT) data with a model that describes the coupled spin dynamics, and (3) inferring the frequency response of neutrons, protons, and electrons to postulated non-magnetic (exotic) interactions. In our analysis, the frequency responses were determined for

each 25-h data segment. The calibration-fit results for Mainz and Kraków are summarized and discussed in the section "Experimental setup."

The DFT is applied to a uniform timeseries of 25-h duration with a rectangular window to generate a DFT subset. To coherently combine ALP DM signals from different DFT subsets, a frequency-dependent shift is applied, and the subsets are phase aligned. Then, all DFT subsets of the respective station are averaged, resulting in a mean value and its standard deviation. We confirmed that the time-shifted DFT averaging procedure had no effect on the Fourier coefficients of injected oscillations.

To search for ALP DM signals [Eq. (10)], we construct an estimator of signal amplitude $S(\omega)$. To achieve the optimum signal-to-noise ratio, we extract all potential ALP DM signatures from the data by combining the Mainz and Kraków DFT datasets. We then combine the power of the carrier and sidebands. Cross-correlated combination of spatially distributed measurements has been performed previously, e.g., ref. 42. In this case specifically, the complex Fourier components of the sidebands ($A_\pm$) in Mainz and Kraków are added with weights taking into account the direction of the sensitive axes and the noise level (see the section "Weights of the ALP signal estimator $S(\omega)$"). The interfered sideband signal then reads

$$A_\pm^{K+M} = \frac{a_\pm^M A_\pm^M + a_\pm^K A_\pm^K\, e^{-i(\phi + \pi/2)}}{a_\pm^K + a_\pm^M} , \quad (11)$$

where $a_\pm^i$ are the weights with $\pm$ designating the higher (+) and lower (−) frequency sideband and superscript $i = M, K$ indicates the Mainz and Kraków stations, respectively. The angle between the positions of the Mainz and Kraków comagnetometers in the rotation plane of the Earth is $\phi$, see Fig. 1b. Accounting for the angle $\phi$ is required to ensure constructive interference of an oscillating ALP DM signal recorded in the two comagnetometers. The constructive interference of the ALP DM signal in the combination of the sidebands is illustrated in Fig. 1c.

Next, we combine the carrier in Kraków $A^K$ with the interfered sideband signal $A_\pm^{K+M}$. The signal estimator $S(\omega)$ for ALP DM incorporates the appropriate weights for averaging signal power; $b^K$ for the carrier in Kraków and $b_\pm^{K+M}$ for the upper and lower interfered sideband, see the section "Weights of the ALP signal estimator $S(\omega)$." The Mainz carrier signal does not contribute to the signal estimator $S(\omega)$, as its weight $b^M \approx 0$, because $\cos(\theta_M) \approx 0$, meaning that there is no contribution to $A^M$ from ALP DM. The estimator for the ALP DM signal $S(\omega)$ is defined in the following way

$$S(\omega) = \sqrt{\frac{b^K|A^K|^2 + b_-^{K+M}|A_-^{K+M}|^2 + b_+^{K+M}|A_+^{K+M}|^2}{b^K + b_-^{K+M} + b_+^{K+M}}} . \quad (12)$$

Note that the standard deviation of the DFTs is propagated accordingly, resulting in an uncertainty in the signal estimator $\Delta S(\omega)$ for each frequency.

The value of $S(\omega)$, given in magnetic-field units and determined from the acquired data, is represented by the blue points in Fig. 2. An ALP DM signal in the $S(\omega)$ data would correspond to a single data point at frequency $\omega_a$. Since this is one point out of the whole dataset consisting of a million points, one per frequency, the distribution of $S(\omega)$ over all frequencies characterizes the technical noise of the interferometer.

In order to estimate the expected value of the technical noise amplitude at each frequency, fit($\omega$), we use the measured values of $S(\omega)$ at surrounding points. For frequencies below 0.1 Hz, the mean is inferred from a global fit assuming a $1/f$ noise model. Above 0.1 Hz the mean is based on the moving average of 500 consecutive points centered around (but excluding) the frequency of interest. Figure 2 shows the mean of the technical noise as an orange line determined as

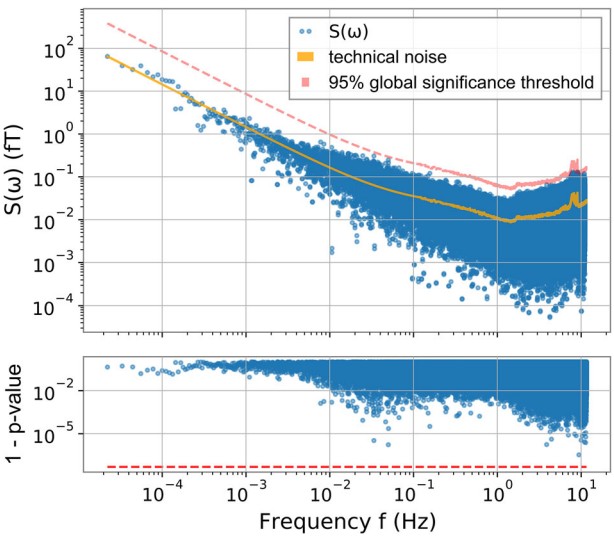

**Fig. 2 | Signal estimator $S(\omega)$, obtained by interfering the Mainz and Kraków comagnetometer data, for frequencies above $\omega_E$.** The results are shown as a function of frequency in the upper figure. The data shows a $1/f$ scaling behavior up to $10^{-2}$ Hz consistent with the technical noise of the apparatus. The peak sensitivity of the estimator reaches $10^{-17}$ T. No ALP candidate is found beyond the global 95% significance threshold in $p$ value, as shown in the lower plot.

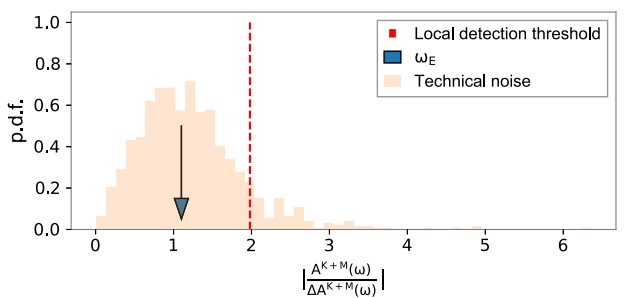

**Fig. 3 | Histogram of the measured-noise distribution and the combined amplitude value at $\omega_E$ as a function of the measured Fourier amplitude.** For an ALP field oscillating below $\omega_E$, the sidebands are not resolved, and an ALP signature is present at $\omega_E$. However, $A^{M+K}(\omega_E)$ is compatible with the expected noise distribution and thus no ALP candidates with frequencies $\omega_a < \omega_E$ are reported. The blue arrow indicates the frequency bin at $\omega_E$, the dashed red line indicates the 95% confidence detection threshold, and the orange distribution is the measured noise for the frequency bins between $\omega_E$ and 0.01 Hz. The amplitude is normalized by the standard deviation $\Delta A^{M+K}(\omega)$ to whiten the noise.

### Table 1 | Summary of constants used in the data analysis

| Symbol | Name | Value | Source |
|---|---|---|---|
| $\xi_N$ | Neutron spin content in ³He | 0.87 | 40 |
| $\xi_P$ | Proton spin content in ³He | 0.027 | 40 |
| $\gamma_N$ | Gyromagnetic ratio of ³He | −32.43 MHz/T | 54 |
| $\gamma_e$ | Effective gyromagnetic ratio of K | −5.38 GHz/T | 55 |
| $\rho_{DM}$ | Local DM energy density | 0.4 GeV/cm³ | 33 |
| $v_O$ | Virial DM velocity | 220 km/s | |
| $v_E$ | Velocity of Earth in the galactic rest frame | 233 km/s | |
| $\omega_E/2\pi$ | Earth sidereal frequency | 11.6 μHz | |
| $\theta_M$ | Mainz sensor polar angle | 90 ± 1° | |
| $\theta_K$ | Kraków sensor polar angle | 50 ± 1° | |

Effective gyromagnetic ratio of K atoms $\gamma_e$ estimated for ³⁹K ($I = 3/2$) in SERF regime, taking into account the slowing down factor polarization level of 50%[55]. We denote the polar angle as the angle of the sensitive axis relative to the rotation axis of the Earth.

resolved in our data. This is due to the fact that the Earth's rotation is modulating the ALP field that would be measured in the experiment, so there would always be a signal oscillating at Earth's sidereal frequency $\omega_E$. Similar analysis techniques have been proposed previously[43–45] that did not take into account the stochasticity of the ALP field. The width of each frequency bin is approximately equal to $\omega_E$. In this situation, the sidebands are at $\omega_E \pm \omega_a$. Since $\omega_E > \omega_a$, they appear in the same frequency bin at $\omega_E$. By examining the frequency bin at $\omega_E$, the ALP search can be extended to the limit where the ALP oscillation is much slower than the rotation of Earth ($\omega_a \ll \omega_E$). Due to the particular characteristics of this bin, we do not include it in the general ALP DM search described in the previous paragraph. Instead, we add Mainz and Kraków amplitudes with their respective weights and obtain $A^{K+M}(\omega)$ following Eq. (11). We use the estimator $A^{K+M}(\omega)/\Delta A^{K+M}(\omega)$ normalized by the standard deviation $\Delta A^{M+K}(w)$ as the expected noise model. For the frequency bins below 0.1 Hz the estimator histogram follows a Rayleigh distribution, which is used to calculate the detection threshold (95% local significance threshold) for the measured value $A^{K+M}(\omega_E)$. The results are shown in Fig. 3. The measured amplitude at $\omega_E$ is $S(\omega_E) = 4.7 \times 10^{-14}$ T, and the standard deviation is $\Delta S(\omega_E) = 4.2 \times 10^{-14}$ T. The standard deviation is calculated from the value of the Fourier coefficients at $\omega$ of each of the 25-h segments. Then, it is propagated following the definition of $S(\omega)$. It is consistent with noise and therefore shows no evidence of an ALP DM candidate.

### Setting limits

Since our measurements did not reveal any ALP DM candidates in the frequency range below 11.6 Hz, we proceed to set limits on the ALP DM pseudoscalar couplings. Specifically, we set exclusions on the ALP-proton coupling $g_{aPP}$, ALP-neutron coupling $g_{aNN}$, and ALP-electron-coupling $g_{aee}$. For simplicity, when constraining one of the three couplings, we do not consider interactions due to the other couplings. We describe the exclusion strategy for a generic $g_{eff}$ coupling to the spin. For proton and neutron couplings, $g_{eff}$ is rescaled by the respective nucleon contributions $\xi_P$ and $\xi_N$ to the total nuclear spin to get the final exclusions for $g_{aPP} = g_{eff}/\xi_P$ and $g_{aNN} = g_{eff}/\xi_N$. The values for ³He of the nuclear content are $\xi_N$ and $\xi_P$ are calculated by using by using the Full-scale shell model, according to ref. 40. Table 1 shows the values for the nuclear contents and the gyromagnetic ratios $\gamma_N$ and $\gamma_e$, among other relevant parameters.

The electron-coupling frequency response of the setup depends on the effect of the magnetic shield. While the effect of the shield on

discussed above, as well as a light red line indicating the 95% global significance threshold for each frequency.

To determine whether any of the estimator data points are significantly larger than expected from the measured technical noise, a detection threshold is established. The set of the signal-estimator values $S(\omega)$, normalized with the expected noise at each frequency, is found to match a non-central $\chi^2$ distribution with six degrees of freedom (see the section "Noise distribution and signal insertion demonstration"). The fitted non-central $\chi^2$ distribution is used to set the detection threshold to guarantee that a candidate signal has only a 5% global chance of arising due to technical noise, which accounts for the look-elsewhere effect. The measured values of $S(\omega)$ are consistent with noise and therefore show no evidence of an ALP DM candidate.

For ALP DM with frequencies below $\omega_E$, a different search approach is used. An ALP DM contribution at $\omega_E$ would be observed for the Compton frequencies $\omega_a < \omega_E$ even if $\omega_a$ cannot be directly

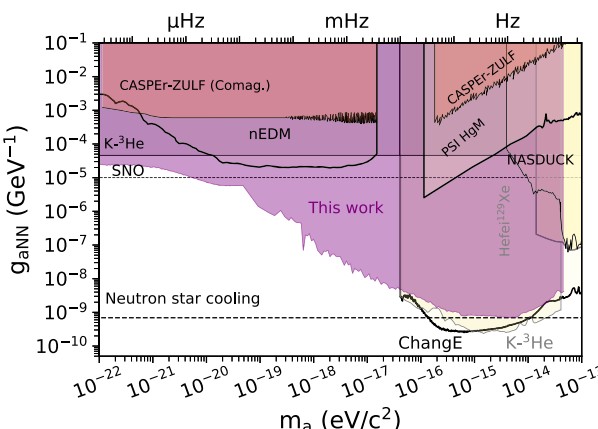

**Fig. 4 | Exclusion plot for the neutron coupling (mean limits).** Other laboratory (solid lines) and astrophysical (dashed lines) constraints are shown for reference and extracted from ref. 56: CASPEr-ZULF[44,53], K-$^3$He[30,51], nEDM[57], PSI HgM[58], SNO[59], NASDUCK[60], Hefei $^{129}$Xe[61], ChangE[31], and neutron-star cooling[62].

the ALP field itself can be neglected, the electron spins in the ferromagnetic shield "see" the exotic field effectively as a weak magnetic field. The shield generates an actual magnetic field in its inner volume to compensate. Therefore, within the shielded volume, we have both the ALP field and the real magnetic field produced by the shield[46].

The frequency-dependent response of the comagnetometer to this combination of fields is discussed in the section "Comagnetometer response to exotic electron interaction." Below, the Larmor precession frequency of $^3$He ($\approx$3 Hz), the comagnetometer is, by design, insensitive to magnetic fields and its frequency response to the ALP field[29] is the same as the nuclear frequency response but rescaled by the gyromagnetic ratio of the electron $g_{aee} = g_{eff}\gamma_N/\gamma_e$.

To compute upper limits on the ALP-coupling strength, the statistical properties of the technical noise distribution and the distribution of ALP DM signatures in the signal estimator $S(\omega)$ have to be considered. The ALP DM signal distribution $S_{g_{eff}}(\omega)$, given an effective coupling $g_{eff}$, is obtained by performing Monte Carlo simulations based on Eq. (12). It accounts for the stochastic properties of ALP DM (section "Signal model") according to Eq. (9).

We use the Confidence Levels (CLs) method[47], commonly used in new particle searches since ref. 48 to determine the limits on the coupling strength $g_{eff}$. If the measured value $S(\omega)$ is purely due to noise, it can be represented by a random number $X_n$ drawn from a non-central $\chi^2$ distribution with a scale parameter determined by the expected value based on the neighboring points (1/$f$ fit for frequencies below 0.1 Hz and moving average for frequencies above). The potential contribution to $S(\omega)$ from an ALP DM signal can also be represented by a random number accounting for its stochastic nature. We compute this with Monte Carlo simulations and scale the ALP DM distribution with the interaction strength $g_{eff}$. Finally, we add the noise distribution to the signal distribution and draw from the combined distribution a random number $X_{S+n}$.

A certain value of $g_{eff}$ is excluded with CL = 95% at frequency $\omega$ when

$$\frac{\mathbb{P}(X_{S+n} \leq S(\omega))}{\mathbb{P}(X_n \leq S(\omega))} \leq 1 - \text{CL} = 0.05, \tag{13}$$

where $\mathbb{P}$ indicates the probability. The condition described by Eq. (13) means that for the excluded value of $g_{eff}$, there is only a 5% relative probability that an ALP DM signal contributes to the measured value of $S(\omega)$.

The sidereal frequency bin at $\omega_E$ is used to expand the search to frequencies below $10^{-5}$ Hz. Due to constraints on ALP DM density

below $m_a = 10^{-22}$ eV/c$^2$[12], the exclusion from the sidereal frequency bin is limited to the range from $2.4 \times 10^{-8}$ Hz up to the sidereal frequency $\omega_E/2\pi \approx 1.1 \times 10^{-5}$ Hz. In this range, the field is considered nearly constant ($\omega_a \approx 0$). The derivation of the signal model in this regime, as well as for the intermediate regime, is shown in the section "Derivation of the ALP signal in the ultra-low oscillating regime $\omega_a \ll \omega_E$."

For frequencies around $10^{-1}$ Hz, the total measurement time is on the order of the ALP field coherence time $\tau_{\omega_a}$. The ALP oscillation is coherent for measurement times $T \lesssim \tau_{\omega_a}/2.5$[36,41]. In particular, for frequencies $w_a/2\pi \gtrsim 5 \times 10^{-2}$ Hz the ALP field is not coherent over the entire measurement time ($T = 92$ days). Therefore, for frequencies above $5 \times 10^{-2}$ Hz, we weaken our estimated constraints on the coupling constants by $\sqrt{T/\tau_{\omega_a}}$ to account for the incoherent averaging, as discussed in ref. 11. We also consider that over several coherence times the stochastic amplitudes [$\alpha_x$, $\alpha_y$, and $\alpha_z$ in Eq. 9] will change. When calculating $S_{g_{eff}}(\omega)$ with MC simulations, we average the stochastic amplitudes over the number of coherence times during the total measurement time.

The final results of this work are constraints on proton, neutron, and electron ALP couplings $g_{aNN}$, $g_{aPP}$, and $g_{aee}$, respectively. The excluded parameter space covers a frequency range from $10^{-8}$ to 11.6 Hz, corresponding to masses from $10^{-22}$ to $4 \times 10^{-14}$ eV, a total of nine orders of magnitude. Figs. 4–6 show the constrained parameter space in the context of previous laboratory searches. We plot the constraints smoothed with a moving average to guide the eye (mean limits). The scatter of the constraint data is similar to that in Fig. 2. In the mass range between $1.2 \times 10^{-17}$ and $4 \times 10^{-17}$ eV, the exclusion improves previous constraints by 3–4 orders of magnitude in $g_{aNN}$ and $g_{aPP}$. The constraints on $g_{aee}$ improve direct DM search constraint by up to one order of magnitude and confirm the exclusions from solar axions searches and stellar physics.

Note that we do not include the results of ref. 45 in Figs. 4 and 6. Reference[45] presented a re-analysis of the comagnetometer data from the Princeton group acquired in three different experiments[49–51]. More recently, the Princeton group published their own re-analysis of their data[30], which notes critical issues in the interpretation of their data not accounted for in ref. 45. We regard the Princeton analysis as the definitive interpretation of the data.

Both comagnetometers in Mainz and Kraków are part of the Advanced GNOME experiment[52]. With additional comagnetometers currently in development, the network is set to expand, significantly enhancing its sensitivity to both ALP DM and transient events in future science runs.

At the same time, this experiment is part of the CASPEr family of experiments[44,53], significantly improving previous CASPEr results (up to seven orders of magnitude) while also extending the covered mass range, see Figs. 4 and 5.

## Methods
### Experimental setup
The interferometer is composed of two self-compensating comagnetometers located about 1000 km apart: one in Mainz, Germany, and the other in Kraków, Poland. The two self-compensating comagnetometers are similar to that reported in refs. 27,29. At the core of the Mainz (Kraków) comagnetometer system is a spherical cell heated to about 180 °C and mounted inside a four-layer magnetic shield. The cell is filled with 3 amg of $^3$He and 50 Torr of $N_2$ and loaded with a drop of an alkali-metal mixture with 1% $^{87}$Rb and 99% natural-abundance K (molar fractions). Spins are optically pumped with a 30 mW/cm$^2$ (50 mW/cm$^2$) circularly-polarized light tuned to the center of the Rb D$_1$ (D$_2$) line. The readout is realized by monitoring the polarization rotation of a -15 mW/cm$^2$ (1 mW/cm$^2$) linearly-polarized light detuned about 0.5 nm from the K D$_1$ line. To reduce the influence of the magnetic-field noise at low frequencies, the comagnetometers are operated in the self-

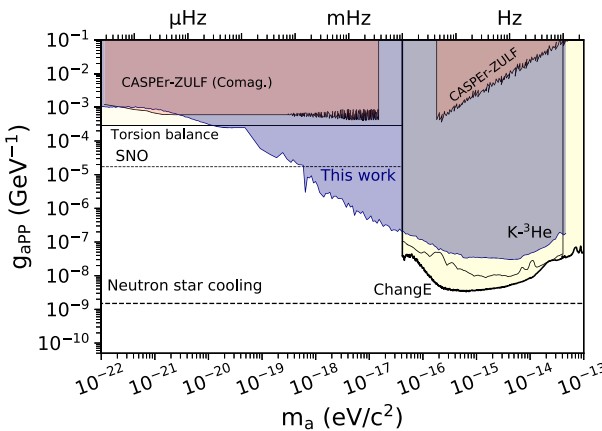

**Fig. 5 | Exclusion plot for the proton coupling (mean limits).** Other laboratory (solid lines) and astrophysical (dashed lines) constraints are shown for reference and extracted from ref. 56: CASPEr-ZULF[44,53], ChangE[31], K-$^3$He (rescaled from $g_{aNN}$ constraints)[30] and neutron-star cooling[62].

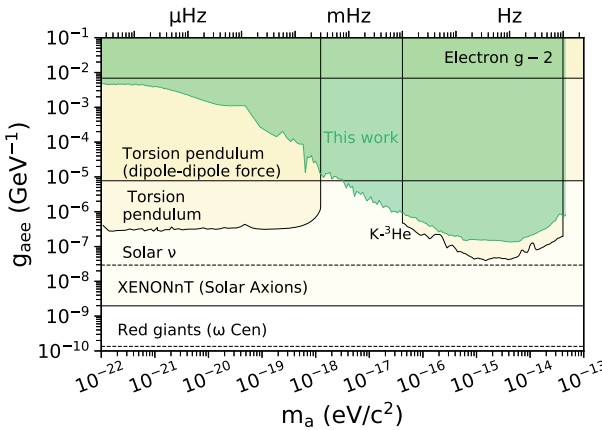

**Fig. 6 | Exclusion plot for the electron coupling (mean limits).** Other laboratory (solid lines) and astrophysical (dashed lines) constraints are shown for reference and extracted from ref. 56: Electron $g$−2[63], Torsion pendulum[64,65], K-$^3$He[30], XENONnT (Solar Axions)[66], Solar neutrinos[67], and red giant branch[68].

compensating regime[25]. To operate in this regime, a $B_z$ (compensation) field of about 100 nT (50 nT) is applied in Mainz (Kraków). In the Mainz comagnetometer, we modulate the $B_x$ field with an 80 Hz sine wave. The signal demodulated at that frequency exhibits a resonance that is used to lock the system to the compensation point and therefore follows slow drifts of the equilibrium compensation field[27]. The sensitivity of both comagnetometers to exotic nucleon couplings is estimated with a daily (every 25 h) calibration pulse according to the calibration procedure described in ref. 29. In the Kraków comagnetometer, the drifts from the compensation point are corrected depending on demand after applying the calibration pulse (see, for example, two groups of fitted values of $\omega_e$ and $\omega_{He}$ in Fig. 7, corresponding to two time-separated datasets), while there is no such a correction procedure in Mainz. The response to the calibration pulse is fitted with such parameters as the amplitude of the response that relates the voltage output to pseudomagnetic fields, the detuning from the compensation point $\Delta B_z$, the Larmor frequency of both electron spin $\omega_e$, and $^3$He nuclear spin $\omega_n$ and the relaxation rate of electrons $R_e$ (at the current stage of the work, nuclear relaxation $R_n$ is ignored). Figure 7 summarizes the fit parameters for all 25 h datasets. As shown, there is little variance in the parameters during the whole run. The detuning from the compensation point, $\Delta B_z$ remains below 5 nT, which is within 5% (10%) of the leading field ~100 nT (~50 nT) in Mainz (Kraków). The fitted parameters encode the effects of ambient drifts of

the environment, such as temperature and external magnetic field, that affect the comagnetometer sensitivity. By recalibrating every 25 h, the effects of such drifts on the low-frequency noise floor of the comagnetometer are taken into account. A detailed discussion of the setup stability on shorter time scales is provided in Chapter 6 of ref. 16. The low-frequency calibration factor for Mainz and Kraków is displayed in Fig. 7.

### Weights of the ALP signal estimator $S(\omega)$

The weights used in the signal estimator [see Eqs. (11) and (12)] are defined in the following way

$$
\begin{aligned}
a^M_\pm &= \frac{\sin\theta_M}{\left(\sigma^M_{A_\pm}\right)^2}, \\
a^K_\pm &= \frac{\sin\theta_K}{\left(\sigma^K_{A_\pm}\right)^2}, \\
b^M &= \frac{2\cos\theta_M}{\left(\sigma^M_{|A|^2}\right)^2} \approx 0, \\
b^K &= \frac{2\cos\theta_K}{\left(\sigma^K_{|A|^2}\right)^2}, \\
b^{K+M}_\pm &= \frac{2}{\left(\sigma^{K+M}_{|A_\pm|^2}\right)^2},
\end{aligned}
\qquad (14)
$$

where $a^i_\pm$ and $b^i_\pm$ are the weights with $\pm$ designating the higher (+) and lower (−) frequency sideband, and index $i = M, K, K+M$ indicates the Mainz, Kraków, and combined signal, respectively. The $\sigma_A$ ($\sigma_{|A|^2}$) represents the standard deviation of the Mainz and Kraków amplitudes (power) in the frequency bin of interest. For the interfered sidebands, the weights are resulting from error propagation of $b^{K+M}_\pm$ according to Eq. (11). The factor 2 in $b^M$, $b^K$ and $b^{K+M}_\pm$ is given by the expected ALP DM signal in the carrier being twice the sidebands [Eq. (9)].

### Noise distribution and signal insertion demonstration

In order to derive the global threshold and the limits, it is important to understand the expected distribution of the signal estimator $S(\omega)$. The signal power estimator $S^2(\omega)$ is a sum of six squares of normally distributed numbers: the real and imaginary Fourier coefficients from three frequency bins. However, to compare noise at different frequencies, we normalize ("whiten") each bin by the estimated mean noise, fit($\omega$). Note that the expected value of the noise is proportional to the standard deviation of $S(\omega)$, namely, $\Delta S(\omega)$. The signal estimator excess square $S^2(\omega)/\Delta S^2(\omega)$ follows a non-central $\chi^2$-distribution, defined as $Z = \sum_{n=1}^{N} X_n^2$, where $\{X_n\}$ is a set of normally distributed random numbers with different means and same variance. From propagation of errors, we have that $\Delta S^2(\omega) = 2S(\omega)\Delta S(\omega)$. As $S(\omega)/\Delta S(\omega) = 2S^2(\omega)/\Delta S^2(\omega)$, they both follow the same distribution. This is also the case for $S(\omega)/\text{fit}(\omega) \sim S(\omega)/\Delta S(\omega)$, which is the estimator used to calculate the global threshold and the limits. The fitted non-central $\chi^2$ distribution reproduces the histogram of $S(\omega)/\text{fit}(\omega)$, especially its tail, which is critical for claiming detection, see Fig. 8.

Let us now consider the modification of the distribution induced by an ALP DM signal. Both noise and ALP DM signal are expected to have the same distribution: Gaussian variables in both Fourier quadratures of the raw Fourier spectrum (before combining the Mainz and Kraków carrier and sidebands). We inject the ALP DM signal in the raw Fourier spectrum as a normally distributed random variable, and then the data are combined to obtain the signal estimator $S(\omega)$ [see Eqs. (11) and (12)].

Figure 9 shows the distribution of $S(\omega/2\pi = 11.1\text{ mHz})$ with an injected ALP DM signal for different coupling strengths $g_{\text{eff}}$ computed according to Eq. (9). To sample the stochastic ALP parameter space, we

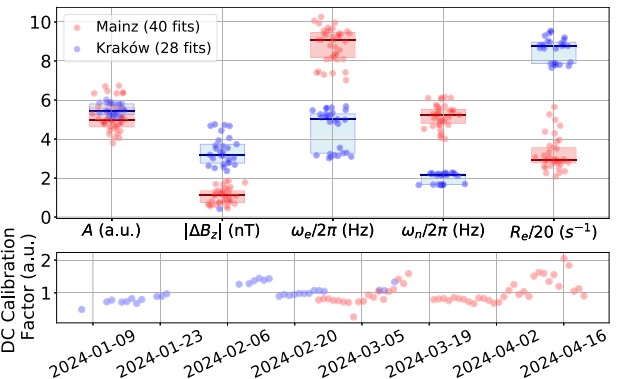

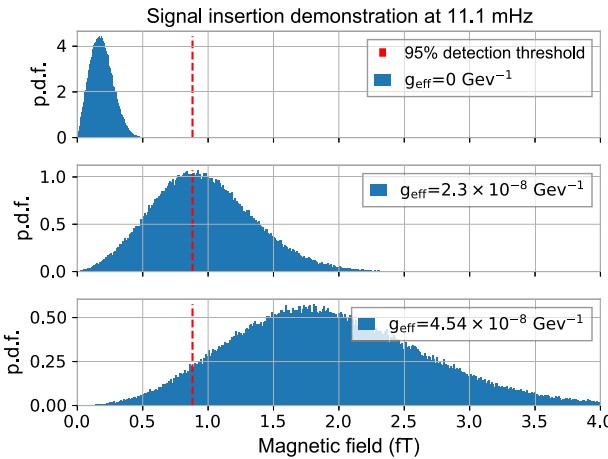

**Fig. 7 | Comagnetometer calibration parameters.** Top: summary of the frequency-response-fit results used for sensor calibration in Mainz and Kraków: response function amplitude $A$, detuning from the compensation point $\Delta B_z$, alkali-metal and noble gas Larmor frequencies $\omega_e$ and $\omega_n$, alkali-metal polarization relaxation rate $R_e$ (divided by 20 only for the better visualization with other parameters). The fit parameters are described in detail in ref. 29. Horizontal bars designate the median values of parameters, and the shaded regions extend to the first quartiles. Bottom: low frequency (DC) calibration factor changes over the measurement timespan.

**Fig. 9 | Insertion of a signal at a frequency bin situated at around 11.1 mHz.** We show the probability distribution function (p.d.f.) of the signal amplitude assuming three different effective couplings $g_{eff}$. The shapes of both noise and signal are the same, since they come from the same distribution. By changing the inserted effective coupling $g_{eff}$, $S(\omega)$ reaches above the detection threshold at this frequency. Due to the measurement time being shorter than the coherence time, this distribution is only sampled once, so it is possible that even for a large coupling strength, the amplitude might still be below the threshold.

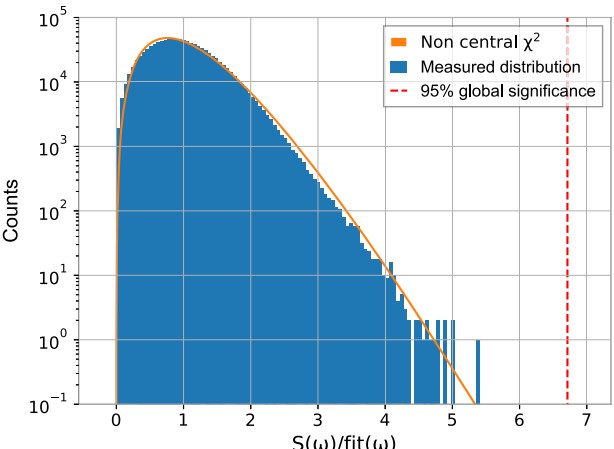

**Fig. 8 | Histogram of the normalized signal estimator $S(\omega)/\text{fit}(\omega)$.** It is obtained by dividing the signal estimator $S(\omega)$ by the expected noise, $\text{fit}(\omega)$. The global threshold at each frequency is determined by fitting the normalized signal estimator with a non-central $\chi^2$ distribution with six degrees of freedom. This enables the calculation of the limits according to Eq. (13). Note that the $y$-axis is on a logarithmic scale to show the tail of the distribution in more detail.

injected $10^6$ different sets of random amplitudes $\alpha_i$ and phases $\phi_i$. The resulting distributions are the sum of the noise and ALP signal distributions. As shown, increasing $g_{eff}$ increases the mean and the variance of the distribution, which has to be taken into account when setting limits.

### Derivation of the ALP signal in the ultra-low oscillating regime $\omega_a \ll \omega_E$

Consider a single station and a quasi-constant oscillating ALP field at $\omega_a \ll \omega_E$. The expected signal, defined in Eq. (8), becomes (ignoring the constant terms)

$$\lim_{\omega_a \to 0} \boldsymbol{\nabla} a(t) \cdot \hat{\boldsymbol{m}}(t) = \sin\theta \{ \alpha_x \cos\phi_x \sin(\omega_E t + \phi_E)$$
$$+ \alpha_y \cos\phi_y \cos(\omega_E t + \phi_E) \}. \quad (15)$$

Therefore, the oscillating signal expected in the sensor is driven only by the rotation of Earth at $\omega_E$ with an amplitude

$$A_{\omega_E} = g_{eff} \sin\theta \sqrt{(\alpha_x \cos\phi_x)^2 + (\alpha_y \cos\phi_y)^2}. \quad (16)$$

In contrast to the case of $\omega_a > \omega_E$ [Eq. (10)], the ALP DM signal amplitude explicitly depends on the random phases $\phi_x$ and $\phi_y$. In this regime, where $\omega_a \approx 0$, there is a probability of measuring near a zero crossing of the oscillation (when $A_{\omega_E} \approx 0$).

We want to account for the reduction of signal amplitude due to sampling a fraction of the oscillation, when $0 < \omega_a < 1/(2\pi T)$, with $T = 92$ days being the total period when the measurements took place. Here, some of the ALP DM oscillation is still measured. We perform Monte Carlo simulations of the integrated power of the respective oscillation fractions. We define this factor as

$$\kappa^2(\omega_a, \phi_x, \phi_y) = \int_0^T dt \, \frac{\alpha_x^2 \cos^2(\omega_a t + \phi_x) + \alpha_y^2 \cos^2(\omega_a t + \phi_y)}{T(\alpha_x^2 + \alpha_y^2)/2}, \quad (17)$$

where $\phi_x$ and $\phi_y$ are uniformly distributed phases between $(0, 2\pi)$. They are sampled in a Monte Carlo procedure to estimate the distribution of $\kappa$ for a given $\omega_a$. The constraints at a frequency $\omega_a < \omega_E$ are given by the product of $\kappa(\omega_a, \phi_x, \phi_y)$ and the signal distribution at $\omega_E$. Then, the limits are calculated by taking a 95% CL with the CLs method.

Figure 10 (left) represents the 95% CL of the factor $\kappa$ depending on the ALP frequency.

### Comagnetometer response to exotic electron interaction

As explicitly shown in ref. 29, the direct coupling of the ALP DM with electrons results in a frequency response that is significantly different from those of neutrons and protons. To accurately estimate the response of the system, one has to consider all possible manifestations of the ALP-electron interaction. The comagnetometer vapor cell is surrounded by a mu-metal shield that cancels the external magnetic fields. However, due to the electron-based mechanism of magnetic shielding, the mu-metal also responds to an exotic electron interaction and induces a "compensating" magnetic field inside the shield (the response of

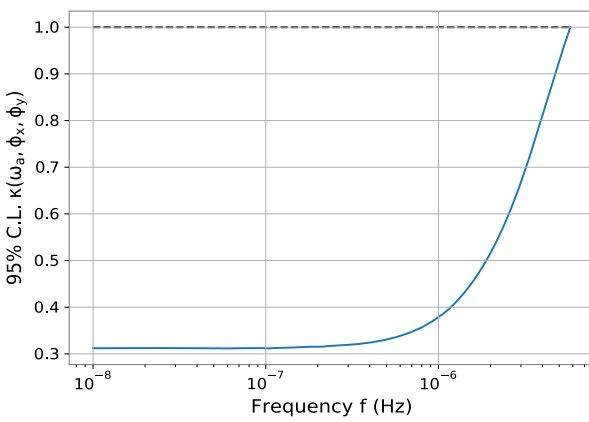
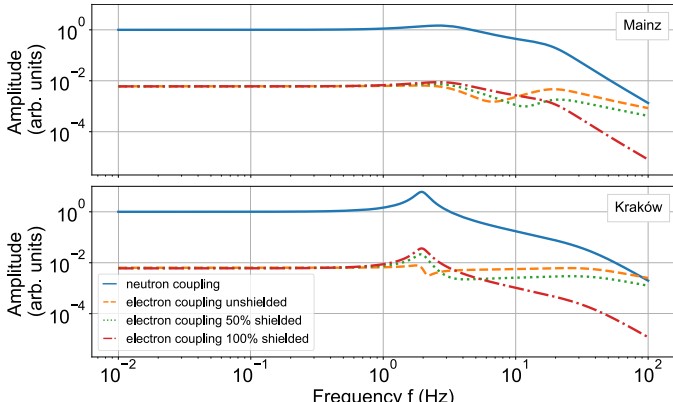

**Fig. 10 | Corrections to the sensitivity of the ALP DM search.** Left: 95% Confidence Level on $\kappa$ in a function of the ALP Compton frequency. The factor $\kappa$ represents the reduction in sensitivity to an ALP oscillation when searching via the Earth's modulation of the field and accounts for the random phase of the ALP field. It also reduces the sensitivity of the set limits accordingly. It reaches a minimum of 0.3. The black dashed line represents a $\kappa$ factor of 1. Right: Comagnetometer frequency response to exotic electronic perturbations. Different shielding scenarios are shown in Mainz (top) and Kraków (bottom): unshielded (dashed), 50% shielded (dotted), fully shielded (dash-dotted). The neutron coupling frequency response (full line) is shown for reference.

the shield in the case of an exotic electron perturbation has been discussed in ref. 46). For the purpose of searching for ALP-electron coupling in the analyzed frequency range, we assume that the ALP-electron shielding factor is of the order of the magnetic shielding factor.

The ALP-electron interaction for a comagnetometer and a mu-metal shield manifests in the following way. The shield generates an opposing real magnetic field as a response to the applied exotic effective field at the sensor position. For a comagnetometer inside the shield, the two fields are superimposed. The electron is influenced by both opposing (magnetic and exotic) fields such that the perturbation of the electron, and consequently its response, is canceled. However, the magnetic field generated by the shield is picked up by the ${}^3\text{He}$ nuclear spins, which do not directly respond to the ALP-electron interaction. Hence, for 100% shielding (i.e., when the induced magnetic field is equal and opposite to the effective exotic field), the frequency response has the same shape as the comagnetometer response to exotic nuclear spin interactions. It is rescaled by the response of the electrons in the shield to the ALP interaction, which is assumed to be flat in the frequency range of interest, effectively attenuating the response by a factor $\gamma_N/\gamma_e$. This is the scenario we assume when setting the limits in Fig. 6.

In general, the nuclear and electron spin perturbation in the comagnetometer is different for arbitrary shielding factors, and then the self-compensating mechanism would not apply. However, an interesting case is when the shield responds only at a 50% level to the exotic electron coupling. In the framework of ref. 28, this corresponds to $\alpha_e = 0.5$ and $\alpha_n = -0.5$. In this case, although the magnetic and exotic interactions are equal in strength, they have opposing directions, and hence the comagnetometer remains sensitive to the electron exotic interaction. The shape of the frequency response is a combination of the electron and nuclear spin responses.

Figure 10 (right) shows the comagnetometer response for three different shielding scenarios. They converge at frequencies below the nuclear Larmor frequency (~3 Hz), since the nuclear spin dominates the dynamics in this regime. Thus, the limits on ALP-electron coupling of this work in this frequency range are almost invariant for any shielding. However, for searches beyond 11.6 Hz, the response changes significantly. For high frequencies (~100 Hz), the unshielded electron response is greater compared to the fully shielded response, and a detailed shielding model should be considered while setting limits. For reference, the nuclear spin response to exotic neutron interactions is also displayed in Fig. 10.

## Data availability
The calibrated data for neutron interactions from Mainz and Kraków comagnetometers are publicly available in https://doi.org/10.6084/m9.figshare.28902860.

## Code availability
The code that support the plots in this paper are available from the corresponding authors upon request.

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

## Acknowledgements

We would like to acknowledge Gary Centers, Wolfgang Gradl, Julian Walter, and Yuzhe Zang for helpful discussions. We acknowledge support by the German Research Foundation (DFG) within the German Excellence Strategy (Project ID 39083149); by work from COST Action COSMIC WISPers CA21106, supported by COST (European Cooperation in Science and Technology). S.P., M.P., and M.S. acknowledge the support from the National Science Center, Poland, within the OPUS program (2020/39/B/ST2/01524), and G.L. acknowledges support from the Excellence Initiative—Research University of the Jagiellonian University in Kraków. The work of D.F.J.K. was supported by the U.S. National Science Foundation under grant PHYS-2110388. The work of A.O.S. was supported by the U.S. National Science Foundation CAREER grant PHY-2145162, and by the Gordon and Betty Moore Foundation, grant DOI 10.37807/gbmf12248.

## Author contributions

D.G.M. and G.L. performed the measurements, analyzed the data, and wrote the manuscript. M.P. and E.K. designed and built the comagnetometer setups in Kraków and Mainz and edited the manuscript. M.S. optimized the comagnetometer in Kraków and performed the calibration and initial steps of the analysis. N.L.F. contributed to data analysis and writing the manuscript. A.O.S. edited the manuscript. D.F.J.K., S.P., and D.B. supervised the research and edited the manuscript. A.W. designed and built the comagnetometer setup in Mainz, conceptualized and supervised research, and wrote the manuscript. All authors contributed with discussions and checking the paper.

## Funding

## Competing interests

The authors declare no competing interests.
