## [Transparent Peer Review file · Nature Communications]

Searching for dark matter with a spin-based interferometer

Corresponding Author: Mr Daniel Gavilan-Martin

Version 0:

Reviewer comments:

Reviewer #1

(Remarks to the Author)

The paper "Searching for dark matter with a 1000 km baseline interferometer" by Gavilan-Martin describes an effort to search for oscillating ALP fields using two atomic comagnetometers. The basic physics is the same as the work in Phys. Rev. X 13, 011050 (2023). However, instead of the single station in the above reference, this work uses the data from two separated stations to coherently search for the oscillating ALP fields. Therefore, the data analysis is more sophisticated, and it also helps to achieve better results in ultra-low frequency range. Overall, the results of this paper seem to be valid and it is an important step forward in this field, which is worth publication in Nature Communications. I would like the authors to consider the following suggestions or questions for the revised paper:

1. According to my understanding, the distance between the two stations is not important for the final results in this work. In other words, I believe that the final results are independent of the distance between the two stations, as long as the sensitive axes remain the same. Therefore, the title of this paper is misleading to stress the word "1000 kilometer baseline". I would suggest the authors to delete these words in the title.
2. The 2nd paragraph of Sec. III describes the data collection information. However, I am still confused about some key information, that is, how many 25-hour continuous data segments are used for the constraints on g_{eff} in this work?
3. Fig. 1 (c) claims to plot all probed frequency bins in both stations. If I understand correctly, there are 1 million frequency bins probed (see Fig. 2, the frequency range covers 10 uHz to 10 Hz with a resolution of 10 uHz). However, it does not look like there are 1 million points in Fig. 1 (c). Am I missing something here?
4. In the end of Sec. V, the standard deviation of $S(\omega_E)$ is mentioned. How is such a standard deviation defined? Is it the standard deviation of an array of $S(\omega_E)$, with each element of $S(\omega_E)$ corresponding to a calculated result from a 25-hour continuous data segment?
5. While the plots of g_{aPP} and g_{aNN} are given, the values of ξ_p and ξ_N , and the nuclear model used for these values are not mentioned. Moreover, this work uses an effectively same system as the K-He3 system in Princeton. Why are the Princeton result only mentioned in the plot of g_{aNN} , not in the plot of g_{aPP} ?

Reviewer #2

(Remarks to the Author)

This paper presents a search for axion-like particle (ALP) dark matter using two atomic comagnetometers located in Germany and Poland. By leveraging the expected spatial and temporal coherence of the ALP field, the experiment probes ALP couplings to neutrons, protons, and electrons across a wide mass range, establishing stringent upper limits on these couplings in previously unexplored parameter spaces. The methodology employed uses the interferometry to access signals from all spatial directions and the exploitation of Earth's rotation representing a notable advancement in dark matter detection techniques. Although the results do not find the existence of ALPs, they strengthen constraints on ALP interactions, contributing meaningfully to the ongoing search for dark matter. I could gladly consider its publication provided that the following questions which need to be adequately addressed.

1. One big technical concern in my eyes is the systematic drift during the long measure (40 days in Mainz and 28 days in Krakow) time in the low frequency region. I would suspect there are parameter drift effects during the data taking. For instance temperature drift, magnetic field drifts. How to characterize these drifts during the measure time? I think a table or a plot of such characterization would be very helpful.

2. This one is related to the first one, I would suspect the sensitivity decrease very fast when going to the low frequency region, consider the fluctuation of the calibration parameters mentioned before. For instance, ΔE has a sharp decrease in low frequency region so they cut the measurement around 10mHz. How can you make sure your results does not decrease a lot due to the parameter drift effects.

3. This work provides a continuous constraint at the lower-mass end of ALPs, a novel achievement compared to existing studies that often have lower-frequency boundaries for constrained ALP masses (e.g., Lee (2023) [Phys. Rev. X 13 (2023) 1, 011050]). Could the authors elaborate on what innovations enable the single frequency bin (ω_E) analysis in this experiment? Specifically, how does the methodology employed here differ from prior work that allows such analysis?

4. In addition to the statistical uncertainties arising from the wavy nature of dark matter, are other uncertainties in the predicted theoretical signal addressed in this paper? For instance, how accurately are the angles $\theta_{K/M}$ maintained and measured throughout the experiment and how would this uncertainty affect Eq.12 ? Furthermore, how do thermal fluctuations of the gas properties, as well as variations in the parameters shown in Fig.7 at each site, impact our ability to predict the ALP signal, and how are such uncertainties included?

5. One of the main purpose of this work, as coherent magnetic field quantum measurements from two distant sites searching for wavy DM, is already well discussed and experimentally accomplished [Nature Commun. 15 (2024) 1, 3331]. One important gain from interference measurements of two distant sites is that one can use cross-correlation to minimize individual noise on each sites, e.g. see Fig.2 in [Nature Commun. 15 (2024) 1, 3331]. Please consider making such analysis, or justify why it is not suitable in this work.

6. The dark matter velocity ($\sim 10^{-3} c$) results in a relatively narrow spectrum. How then does this narrow spectral width lead to the much broader signal distribution (of the order ~ 1) shown in Fig. 9? Clarifying this apparent discrepancy would strengthen the discussion of signal characteristics.

7. The κ -factor in Eq. 17 is somewhat unclear. If I understand correctly, for $\omega_a < \omega_E$, the authors rely on single frequency bin strength data at ω_E from each day, fitting all days' data with a Gaussian and requiring no excess from low-mass ALPs. This should indeed include a suppression effect due to the unknown phases $\phi_{x,y}$. However, why do the authors integrate over time t which is 92 days in Eq. 17 rather than over $\phi_{x,y}$? Neither the stochastic wave perspective (averaging over random $\phi_{x,y}$ distributions from different ALP particles) nor the coherent wave segment perspective (averaging over randomly distributed initial $\phi_{x,y}$) seem to justify this choice.

8. For better clarity and reproducibility, please consider explicitly listing all critical parameters used in this study. For example, parameters such as $\xi_{P/N}$, $\gamma_{N,E}$, and those associated with the $\text{fit}(\omega)$ function would be valuable references for readers and future research.

Specific Comments:

8(minor). The technical noise fit in Fig.2 describes the statistical behavior of data. Is there any way to further explain, locate and calibrate the source of the measured noise?

9. In the paragraph line.171-178, please either define how \hat{x} and \hat{y} are oriented in the chosen coordinate system, or avoid using them directly in the equations without proper clarification.

10. Are there any theoretical or experimental justifications for the specific choice of signal weights between sites and directions as described in Eqs. (11) and (12)? Clarifying this choice would strengthen the argument for the coherence-based analysis and ensure its reproducibility.

11(minor). The symbol ω_n appears first in line 163 to represent the energy of a dark matter particle and later in line 536 to denote the nuclear spin of He3 . Please consider using distinct symbols for these unrelated terms.

12(minor). The labels are hard to identify in Fig.4, especially, ΔE and K-3He.

Reviewer #3

(Remarks to the Author)

Key results:

The manuscript presents the results of an interferometric axion (or ALP) dark matter search spanning a wide mass range — nine orders of magnitude. The experimental apparatus comprises two K-Rb- ^3He atomic comagnetometers separated by 860 km, with one located in Poland and the other in Germany. The authors provide constraints on spin-dependent interactions between axions and protons, neutrons, and electrons. Over large parts of the parameter spaces, the results provide the most constraining laboratory constraints on the axion couplings.

Significance:

Axion dark matter is currently the candidate receiving most attention in dark matter research. As such, leading laboratory constraints are important for the field.

A key challenge for many laboratory-based axion searches is their reliance on resonant conversion, which makes constraining large portions of parameter space difficult. The approach described in this manuscript is interesting because it is broadband and can constrain large mass ranges. The authors improve upon existing laboratory constraints by many

orders of magnitude, although their constraints remain below those established through astrophysical observations.

A point that was not fully clear to me: It was mentioned that comagnetometers have already established stringent limits in certain mass ranges. I was not sure if the results of this work constitute the first constraints from an interferometer composed of two comagnetometers.

Validity, Methodology and Data Assessment:

Through their past publication record, the authors are experts in the field. They explain their methods clearly and concisely and, when appropriate, refer to their earlier papers.

Overall, I believe the approach taken is valid. There is a robust discussion of the signal model and subtleties in modelling the axion field depending on whether the axion's coherence time is longer or shorter than the measurement time. The authors provide a good discussion of the appearance of three frequency components owing to the rotation of the Earth, and how the comagnetometers in Poland and Germany provide complementary information owing to their different orientations. The signal estimator used to compare measurements with theory is also well motivated.

The methods section provides sufficient detail for the work to be reproduced. For the analysis components, the authors describe their approach clearly, though someone would need to run their own Monte Carlo simulations to fully replicate the results. This is reasonable given the nature of the work.

I have more familiarity with the interpretation side of this research and feel more confident evaluating those aspects of the manuscript. Specific aspects of the experimental setup and data processing techniques fall outside my core expertise, though I was able to follow the overall methodology.

The figures are clear and effective. Figure 1 provides a good representation of the methodology. The remaining figures present measurements, interpretations, and comparisons with other constraints thoroughly. The constraints from other experiments are clearly represented.

The text is well written with appropriate referencing. The manuscript effectively cites the relevant literature on axion searches, comagnetometer techniques, and previous constraints. The authors provide a concise and complete description of their work.

Suggested Improvements:

The following specific points would improve clarity for readers:

1. The title emphasises the long baseline of the interferometer. However, this seems less relevant in the actual analysis since the approximation $k \cdot r = 0$ is used. Perhaps it would be better to emphasise the different orientations of the comagnetometers rather than their separation distance.
2. For non-expert readers, the authors should state explicitly that they don't account for the dark matter escape speed in equation 3. This is a reasonable approximation, but it should be noted as such.
3. It's not clear if ω_a was defined separately in the manuscript.
4. Regarding the DFT method: The authors should clarify whether the time series is uniform and if they apply any window functions to their DFT subsets.
5. Since both ω and f are referred to as frequency, I suggest adding a symbol, either f or $\omega/(2\pi)$, to the frequency scale in Figure 2 to make it immediately clear to the reader what is being plotted.
6. The statement about assuming a $1/f$ technical noise model below 0.1 Hz is somewhat unclear. Perhaps they meant 0.01 Hz, as it seems to depart from $1/f$ between 0.01 and 0.1 Hz.
7. At the beginning of Section IV, the statement that they don't see any axion signals from 10^{-8} Hz to 11.6 Hz is initially confusing. It later becomes clearer that they stop their interpretation at that frequency because of constraints on axions making up all dark matter. I suggest they drop the initial statement to the frequency range since it is explained more carefully after equation 13.
8. Could the authors comment on whether the CLs method standard for this type of analysis?

Version 1:

Reviewer comments:

Reviewer #1

(Remarks to the Author)

The authors have answered all my questions and made corresponding changes. I have no other comments, and would like to recommend the paper to be accepted for publication in Nature Communications.

Reviewer #2

(Remarks to the Author)

I would like to thank the authors for their detailed responses and careful revisions. The authors have addressed the majority of my previous comments, and their clarifications are, for the most part, satisfactory. I appreciate their efforts in improving the manuscript.

At this stage, I believe the paper meets the standards for publication in Nature Communications, and I recommend its acceptance.

That said, I still have a few remaining questions and suggestions that the authors may wish to consider in the final revision:

1. There are a lot of constraint lines and names in Fig.4, which make it hard to identify. My previous thought was to suggest the authors to use colors, line styles, etc. to improve its readability. The font size is not critical but I still appreciate the authors' effort.
2. It is nice for the authors to show the calibration results on each 25 h-span. However based on the data points they show, it seems reasonable to question if one would expect a factor of 2 drifting at the time scale of a day. Thus at this work's critical frequency ω_E , a large uncertainty due to a not-perfect measurement equipment itself should be expected. It would be really helpful if the authors can elaborate on either how such effects are already included in the noise spectrum or why doesn't it contribute significantly to the results.
3. I want to thank the authors for explaining Fig.9 and also Eq.(17). It seems that they are both accounting for the Monte Carlo process characterizing the stochastic behavior of axion background. It is still not very clear to me though, why it is necessary to implement Monte Carlo to determine the spatial derivative and phase status of the axion background, which seems to be one of the innovation point of this paper. It can be helpful if a line representing a standard virialized, isotropically distributed, evenly phase-distributed axion background can be drawn in a relevant plot, showing the advantage of Monte Carlo method.
4. It seems that there would be a most conservative constraint, by assuming the axion background is at an extreme phase where the theoretically induced signal would be smallest, i.e. suppressed by $m_a T (=92 \text{ days})$. The authors seem to acknowledge such existence and take a Monte Carlo method which is already beyond the standard of this field. However it can be interesting to know where the most conservative constraint would be.

I hope the authors can take a final look at these issues, though none of them are critical enough to delay publication.

Reviewer #3

(Remarks to the Author)

The authors have satisfactorily addressed my list of suggested improvements in the updated version of the manuscript. After incorporating the comments from all referees, the clarity of the paper is much improved.

However, there is still one point from my previous report that remains. This was within the main text rather than a separate bullet point, so apologies if it was not clearly highlighted in my previous report. Copying from my previous report, the point that remains unclear is: It was mentioned that comagnetometers have already established stringent limits in certain mass ranges. I was not sure if the results of this work constitute the first constraints from an interferometer composed of two comagnetometers.

Let me clarify further. This concern relates to the paper's novelty and positioning within the field. In particular, I believe the introduction (or abstract) could more clearly articulate: the specific gap in existing knowledge this work addresses; how this approach differs from previous comagnetometer studies (e.g., compared to the K-3He systems shown in Figs 4 and 5); the advantages this new approach offers compared to existing methods.

I believe this change would enhance both the impact of the paper and readers' understanding of its importance in the field.

Title: Searching for dark matter with a **spin-based** interferometer

Authors: Daniel Gavilan-Martin, Grzegorz Lukaszewicz, Mikhail Padniuk, Emmanuel Klinger, Magdalena Smolis, Nathaniel L. Figueroa, Derek F. Jackson Kimball, Alexander Sushkov, Szymon Pustelny, Dmitry Budker, Arne Wickenbrock

Dear Referees,

We thank you for the careful reading of the manuscript and for providing detailed comments and suggestions. Below, we list the comments, item by item, followed by our reply and action taken.

First Referee

Comment 1:

According to my understanding, the distance between the two stations is not important for the final results in this work. In other words, I believe that the final results are independent of the distance between the two stations, as long as the sensitive axes remain the same. Therefore, the title of this paper is misleading to stress the word “1000 kilometer baseline”. I would suggest the authors to delete these words in the title.

Our reply:

Thank you for your comment. There is no sensitivity scaling with the distance. We get an improvement in sensitivity by the fact that each comagnetometer is in a different geographical position and with different orientation. We understand that this could be misleading, especially to those more familiar with LIGO and other gravitational wave detectors where the sensitivity depends directly on the distance.

Action taken: We have changed the title to “**Searching for dark matter with a spin-based interferometer**”

Comment 2:

The 2nd paragraph of Sec. III describes the data collection information. However, I am still confused about some key information, that is, how many 25-hour continuous data segments are used for the constraints on g_{eff} in this work?

Our reply:

Thank you for noticing the mistake. We inaccurately used “days” instead of “25-hour data segments”. In total, there are 40 25-hour data segments in Mainz and 28 in Kraków.

Action taken: We have changed days to 25-hour segments in section III (line 255): “**The data analyzed in this work correspond to a total of 40 25-hour segments in Mainz and 28 25-hour segments in Kraków**” and changed the label in Fig. 7 from “days” to “fits”.

Comment 3:

Fig. 1 (c) claims to plot all probed frequency bins in both stations. If I understand correctly, there are 1 million frequency bins probed (see Fig. 2, the frequency range

covers 10 uHz to 10 Hz with a resolution of 10 uHz). However, it does not look like there are 1 million points in Fig. 1 (c). Am I missing something here?

Our reply:

We thank the reviewer for the comment. Fig. 1c) is an illustration, so the total number of points is not the same as the experimental data in Sec. III.

Action taken:

In the caption of Fig. 1 c) we have added: **“The frequency points are intended for illustration purposes and do not correspond to experimental data.”**

Comment 4:

In the end of Sec. V, the standard deviation of $S(\omega_E)$ is mentioned. How is such a standard deviation defined? Is it the standard deviation of an array of $S(\omega_E)$, with each element of $S(\omega_E)$ corresponds to a calculated result from a 25-hour continuous data segment?

Our reply:

We agree with the reviewer that the standard deviation of $S(\omega_E)$ might not be clear enough. It is derived from the standard deviation of the real and imaginary parts of the Fourier coefficients at frequency ω of the array of all the 25 hour segments. Then, the error is propagated according to standard propagation of errors following $S(\omega)$ definition.

Action taken:

At the end of Sec. III (line 387), we have included: **“The standard deviation is calculated from the value of the Fourier coefficients at ω of each of the 25-hour segments. Then, it is propagated following the definition of $S(\omega)$.”**

Comment 5:

While the plots of g_{aPP} and g_{aNN} are given, the values of ξ_p and ξ_N , and the nuclear model used for these values are not mentioned. Moreover, this work uses an effectively same system as the K-He3 system in Princeton. Why are the Princeton result only mentioned in the plot of g_{aNN} , not in the plot of g_{aPP} ?

Our reply:

We are using the values for Helium 3 of the Full-scale model that can be found in Ref. 39. By using this model, one can rescaled the limits of the Princeton group and also constraint g_{aPP} , since they also use Helium 3.

Action taken:

We have added to the beginning of Sec.IV (line 406): **“The values for ${}^3\text{He}$ of the nuclear content are ξ_N and ξ_P are calculated by using by using the Full-scale shell model, according to Ref.\, \cite{kimball_nuclear_2015}. Table 1 shows the values for the nuclear contents and the gyromagnetic ratios γ_N and γ_e , among**

other relevant parameters.” We have also added a table that summarizes some of the experimental parameters.

We have included the rescaled constraints of the Princeton group in g_aPP in Fig. 5 and add to its caption: **“K-3He (rescaled from g_{aNN} constraints)”**

Second Referee

Comment 1, 2 and 9:

One big technical concern in my eyes is the systematic drift during the long measure (40 days in Mainz and 28 days in Krakow) time in the low frequency region. I would suspect there are parameter drift effects during the data taking. For instance temperature drift, magnetic field drifts. How to characterize these drifts during the measured time? I think a table or a plot of such characterization would be very helpful.

This one is related to the first one, I would suspect the sensitivity decrease very fast when going to the low frequency region, consider the fluctuation of the calibration parameters mentioned before. For instance, ChangeE has a sharp decrease in low frequency region so they cut the measurement around 10mHz. How can you make sure your results does not decrease a lot due to the parameter drift effects.

The technical noise fit in Fig.2 describes the statistical behavior of data. Is there any way to further explain, locate and calibrate the source of the measured noise?

Our reply:

Thank you for your comments. There are indeed parameter drifts over the course of such a long measurement. To reduce its effect and take them into account, we have done two things.

First, in the Mainz station a PID was designed to keep the comagnetometer operating at the compensation point during the whole measurement (as mentioned in Sec. V a)). Second, every 25-hour segment is calibrated independently with a calibration pulse of 10 pT that is used to determine our frequency-dependent sensitivity. With these pulses, we can estimate all relevant parameters of our system and assess if the system is still maximally sensitive or if the ambient drifts or any other effect has reduced the sensitivity. These parameters are summarized in Fig. 7. The fluctuation of these parameters are reflected in the sensitivity conversion of our signal to magnetic field units and they manifest in the calculation of the noise floor in Fig. 2.

With respect to the measured noise, in Sec. V c) we show that our noise follows a non-central chi distribution resulting from the combination of frequency bins of both stations. We show in Fig. 8 that this distribution indeed fits our data, including a very good reproduction of the tail, critical for claiming a hypothetical detection.

In general, it is highly non-trivial to estimate the error budget and origin of the noise sources. Instead, these noises are taken into account by assessing the sensitivity with calibration pulses, with no specific investigation of their origin. Nonetheless, we are of course interested in understanding what sources of noise are most relevant to us. We suspect after performing some more measurements after this campaign that we may be limited mainly by magnetic noise, including geo-magnetic and human-made components and more. However, we consider that this discussion is beyond the scope of our work, and

that it is sufficient to analytically understand the nature of the statistics of our noise, as it is described in Sec. IV c).

Action taken:

We have added in Sec. V a) the following sentences (line 587): **“The fitted parameters encode the effects of ambient drifts of the environment, such as temperature and external magnetic field, that affect the comagnetometer sensitivity. By recalibrating every 25 h the effects of such drifts on the low-frequency noise floor of the comagnetometer are taken into account. The low-frequency calibration factor for Mainz and Kraków are displayed in Fig. 7”**

We have also modified Fig. 7 by adding calibration factor changes over time in the bottom plot. The caption of Fig. 7 is adjusted accordingly: **“Top: (...). Bottom: low frequency (DC) calibration factor changes over the measurement timespan.”**

Comment 3:

This work provides a continuous constraint at the lower-mass end of ALPs, a novel achievement compared to existing studies that often have lower-frequency boundaries for constrained ALP masses (e.g., Lee (2023) [Phys. Rev. X 13 (2023) 1, 011050]). Could the authors elaborate on what innovations enable the single frequency bin (ω_E) analysis in this experiment? Specifically, how does the methodology employed here differ from prior work that allows such analysis?

Our reply:

Thank you for your comment. This is not really an innovation, as some previous work have proposed the idea to extend the constraints to arbitrarily low frequencies by using the daily modulation signature (Bloch, 2020; Graham, 2018; Wu, 2019). In our case, the fact that we measure continuously for segments of 25 hours allows us to resolve the sidereal frequency of the Earth. The ALP field measured in the experiment would be modulated, so a signal at the Earth sidereal frequency is always present, even in the limit of a DC ALP field. Any other experiment with a sensitive axis, such as the Princeton group experiment, could do the same if they measure long enough.

Bloch, I.M., Hochberg, Y., Kuflik, E. et al., *Axion-like relics: new constraints from old comagnetometer data*. J. High Energ. Phys. 2020, 167 (2020).

Graham P.W., Kaplan D.E., Mardon J., Rajendran S., Terrano W.A., Trahms L., and Wilkason T., *Spin precession experiments for light axionic dark matter*. Phys. Rev. D 97, 055006 (2018).

Wu, T. et al., *Search for Axionlike Dark Matter with a Liquid-State Nuclear Spin Comagnetometer*. Phys. Rev. Lett. 122, 191302 (2019)

Action taken: We have added a sentence in Sec. III (line 362) when discussing the ω_E analysis: **“This is due to the fact that the Earth rotation is modulating the ALP field that would be measured in the experiment, so there would always be a signal oscillating at Earth sidereal frequency ω_E . Similar analysis techniques have been proposed previously [42, 43, 44] that did not take into account the stochasticity of the ALP field.”**

Comment 4:

In addition to the statistical uncertainties arising from the wavy nature of dark matter, are other uncertainties in the predicted theoretical signal addressed in this paper? For instance, how accurately are the angles $\theta_{K/M}$ maintained and measured throughout the experiment and how would this uncertainty affect Eq.12? Furthermore, how do thermal fluctuations of the gas properties, as well as variations in the parameters shown in Fig.7 at each site, impact our ability to predict the ALP signal, and how are such uncertainties included?

Our reply:

Thank you for your comment. We agree that more details about the measurement of the angles are in order. The angles are measured with a compass, with a precision of 1 degree. The system is screwed tightly in an optical table, and the error during the measurement period is smaller than the precision of the measurement, as this was repeated at the end of the measurement campaign. Compared to the statistical noise in $\Delta S(\omega)$, the angle uncertainty is negligible.

The thermal fluctuations are taken into account in the sensitivity calibration every 25 hours. The fluctuation of the experimental parameters are reflected in the sensitivity conversion of our signal to magnetic field units and they manifest in the calculation of the noise floor in Fig. 2. Additionally, the temperature of the vapor cell is fixed with a PID in both stations, as it is discussed in Ref. 26.

Action taken: We have included the uncertainties when defining the angles at the end of Sec. II (line 241) and added the following sentence: **“The uncertainties on the angle are negligible compared to the statistical noise of the experiment”**

Comment 5 and 11:

One of the main purpose of this work, as coherent magnetic field quantum measurements from two distant sites searching for wavy DM, is already well discussed and experimentally accomplished [Nature Commun. 15 (2024) 1, 3331]. One important gain from interference measurements of two distant sites is that one can use cross-correlation to minimize individual noise on each sites, e.g. see Fig.2 in [Nature Commun. 15 (2024) 1, 3331]. Please consider making such analysis, or justify why it is not suitable in this work.

Are there any theoretical or experimental justifications for the specific choice of signal weights between sites and directions as described in Eqs. (11) and (12)? Clarifying this choice would strengthen the argument for the coherence-based analysis and ensure its reproducibility.

Our reply:

Thank you for your comment. We do in fact follow a similar procedure as the mentioned reference, which is well familiar to us. We add the two stations of the interferometer with weights. The main difference here is that there are three independent traces of the signal (the carrier and the sidebands), and that the orientation of the sensitive axis in the Earth frame is taken into account in the weights, as they modify the signal amplitude that would be observed in each station. The specific values of the weights are discussed in the method section b).

Action taken: We have added a reference to the mentioned work and the following sentence in Sec. III (line 294): **“Cross-correlated combination of spatially distributed measurement has been performed previously, see, for example, Ref. [41]”**

Comment 6:

The dark matter velocity ($\sim 10^{-3} c$) results in a relatively narrow spectrum. How then does this narrow spectral width lead to the much broader signal distribution (of the order ~ 1) shown in Fig. 9? Clarifying this apparent discrepancy would strengthen the discussion of signal characteristics.

Our reply:

The ALP signal has order one variance in the signal distribution when measuring shorter than a coherence time, as we show in section II. Kindly note that Fig. 9 shows the distribution in signal amplitude and not in frequency, so the spread of frequencies is not relevant for this figure.

Action taken:

We have added the following sentence to the caption of Fig. 9: **“We show the probability distribution function (p.d.f.) of the signal amplitude assuming three different effective couplings g_{eff} ”**

Comment 7:

The κ -factor in Eq. 17 is somewhat unclear. If I understand correctly, for $\omega_a < \omega_E$, the authors rely on single frequency bin strength data at ω_E from each day, fitting all days' data with a Gaussian and requiring no excess from low-mass ALPs. This should indeed include a suppression effect due to the unknown phases $\phi_{x,y}$. However, why do the authors integrate over time t which is 92 days in Eq. 17 rather than over $\phi_{x,y}$? Neither the stochastic wave perspective (averaging over random $\phi_{x,y}$ distributions from different ALP particles) nor the coherent wave segment perspective (averaging over randomly distributed initial $\phi_{x,y}$) seem to justify this choice.

Our reply:

Thank you for your comment. The random phases are the relevant parameters. We are interested in the regime where not a full oscillation is measured in the total measurement time and how it affects the signal amplitude. For that purpose, the phases are randomized with a Monte Carlo. Integration over time takes into account the fact that the ALP field evolves in the measurement timespan. By this method we take into account the reduced sensitivity due to the oscillation “being near a zero-crossing”. This enters the estimation in the constraint procedure with the CLs method and we take a 95% C.L. (eq. 13). Additionally, the suppression given by adding incoherently all the ALP waves is given by a Rayleigh distributed number in Eq. 6.

Action taken: We have changed $\kappa(\omega_a)$ to $\kappa(\omega_a, \phi_x, \phi_y)$ to further emphasize the angle dependency in Eq. 17. The section V d) has been edited and

new sentences introduced. We have also realized that κ was missing a square root in the definition. This is also now fixed in eq. 17.

Edits:

(line 645) **“We want to account for the reduction of signal amplitude due to sampling a fraction of the oscillation, when $0 < \omega_a < 1/(2\pi T)$, with $T=92$ days is the total period when the measurements took place.”**

(line 653) **“They are sampled in a Monte Carlo procedure to estimate the distribution of κ for a given ω_a . The constraints at a frequency $\omega_a < \omega_{\text{E}}$ are given by the product of $\kappa(\omega_a, \phi_x, \phi_y)$ and the signal distribution at ω_E . Then, the limits are calculated by taking a 95% C.L. with the CLs method.”**

Comment 8:

For better clarity and reproducibility, please consider explicitly listing all critical parameters used in this study. For example, parameters such as $\xi_{P/N}$, $\gamma_{N,E}$, and those associated with the fit(ω) function would be valuable references for readers and future research.

Our reply:

Thank you for your suggestion. We have added Tab. I with the relevant parameters in the Methods Section.

Action taken:

We have added Tab. I in Sec. V F (line 723) listing all relevant numerical parameters that are necessary for results reproducibility.

Comment 10:

In the paragraph line.171-178, please either define how \hat{x} and \hat{y} are oriented in the chosen coordinate system, or avoid using them directly in the equations without proper clarification.

Our reply:

The orientation of the x and y are not relevant for the following equations as long as the z axis is pointing in the rotation axis of the Earth. However, we agree with the reviewer that it could be misleading, so we have defined it with x in the direction perpendicular to the Greenwich meridian.

Action taken: In Sec. II, when defining the coordinate system, the sentence now reads (line 174): “The coordinate system is chosen such that the z component is parallel to the Earth rotation axis **and the x component is perpendicular to the Greenwich meridian.**”

Comment 12:

The symbol ω_n appears first in line 163 to represent the energy of a dark matter particle and later in line 536 to denote the nuclear spin of He3 . Please consider using distinct symbols for these unrelated terms.

Our reply:

Thank you for noticing. Indeed they are completely unrelated and should have different names.

Action taken:

We have changed the nuclear spin Larmor frequency to “ ω_{He} ” in line 536 (line 554 in the edited version).

Comment 13:

The labels are hard to identify in Fig.4, especially, chanE and K-3He .

Our reply:

We agree that the labels are too small.

Action taken:

We have enlarged the labels in Fig. 4.

Third Referee

Comment 1:

The title emphasises the long baseline of the interferometer. However, this seems less relevant in the actual analysis since the approximation $k \cdot r = 0$ is used. Perhaps it would be better to emphasise the different orientations of the comagnetometers rather than their separation distance.

Our reply:

Thank you for your comment. There is no sensitivity scaling with the distance. We get an improvement in sensitivity by the fact that each comagnetometer is in a different geographical position and with different orientation. We understand that this could be misleading, especially to those more familiar with LIGO and other gravitational wave detectors where the sensitivity depends directly on the distance.

Action taken: We have changed the title to “**Searching for dark matter with a spin-based interferometer**”

Comment 2:

For non-expert readers, the authors should state explicitly that they don't account for the dark matter escape speed in equation 3. This is a reasonable approximation, but it should be noted as such.

Our reply:

Thank you for the comment. We have included a sentence to explicitly mention this approximation

Action taken:

We have added in Sec. II (line 160): **“The dark matter escape velocity is neglected, since it has a negligible effect on the distribution.”**

Comment3:

It's not clear if ω_a was defined separately in the manuscript.

Our reply:

Thank you for noticing. It was not explicitly defined.

Action taken:

We have included in Section II (line 166) the following sentence: **“ ω_a is the Compton angular frequency of the ALP”**

Comment 4:

Regarding the DFT method: The authors should clarify whether the time series is uniform and if they apply any window functions to their DFT subsets.

Our reply:

Thank you for your comment. The timeseries is uniform and we apply a rectangular window.

Action taken:

We have included at the beginning of Sec. III (line 280): **“The DFT is applied to a uniform time series of 25 hours duration with a rectangular window to generate a DFT subset.”**

Comment 5:

Since both ω and f are referred to as frequency, I suggest adding a symbol, either f or $\omega/(2\pi)$, to the frequency scale in Figure 2 to make it immediately clear to the reader what is being plotted.

Our reply:

We agree with the reviewer that it would make the figure clearer.

Action taken:

We have added an “ f ” in the x label of Fig. 2.

Comment 6:

The statement about assuming a $1/f$ technical noise model below 0.1 Hz is somewhat unclear. Perhaps they meant 0.01 Hz, as it seems to depart from $1/f$ between 0.01 and 0.1 Hz.

Our reply:

We tried several fit ranges and we found out that by fitting a $1/f$ with an intercept the quality improves when the frequency range is increased up to 0.1 Hz. However, we agree with the reviewer that the statement in the caption of Figure 2 is misleading since there is a clear change in the slope of the noise floor.

Action taken:

We have changed in caption of Fig. 2 the upper limit of the $1/f$ scaling to be 0.01 Hz: **“The data shows a $1/f$ scaling behaviour up to 10^{-2} Hz consistent with technical noise of the apparatus.”**

Comment 7:

At the beginning of Section IV, the statement that they don't see any axion signals from 10^{-8} Hz to 11.6 Hz is initially confusing. It later becomes clearer that they stop their interpretation at that frequency because of constraints on axions making up all dark matter. I suggest they drop the initial statement to the frequency range since it is explained more carefully after equation 13.

Our reply:

We agree that the lower limit is confusing without further explanation.

Action taken:

We have removed the lower bound in the sentence. It now reads (line 395): “in the frequency range **below** 11.6 Hz”, following the reviewer recommendation.

Comment 8:

Could the authors comment on whether the CLs method standard for this type of analysis?

Our reply:

The CLs method is commonly used for search for new particles in colliders such as CERN and others. It was first used in 2003 when the LEP was searching for the Higgs boson, and has been widely used since then.

Action taken:

We have added in Sec. IV (line 437): “We use the Confidence Levels (CLs) method [43], **commonly used in new particle searches since Ref. [44]**”

We hope that with these corrections, the manuscript can be accepted for publication in Nature Communications.

Sincerely,

Daniel Gavilan-Martin and Grzegorz Lukaszewicz on behalf of the authors

Title: Searching for dark matter with a spin-based interferometer

Authors: Daniel Gavilan-Martin, Grzegorz Lukaszewicz, Mikhail Padniuk, Emmanuel Klinger, Magdalena Smolis, Nathaniel L. Figueroa, Derek F. Jackson Kimball, Alexander Sushkov, Szymon Pustelny, Dmitry Budker, Arne Wickenbrock

Dear Referees,

We thank you again for providing detailed comments and suggestions. Below, we list the comments, item by item, followed by our reply and action taken.

First Referee

We thank the referee for their kind words.

Second Referee

Comment 1:

There are a lot of constraint lines and names in Fig.4, which make it hard to identify. My previous thought was to suggest the authors to use colors, line styles, etc. to improve its readability. The font size is not critical but I still appreciate the authors' effort.

Our reply: We thank the reviewer for his comments. We have changed the color of two limits to gray, hopefully improving the readability.

Action taken: Figure 4 is updated

Comment 2:

It is nice for the authors to show the calibration results on each 25 h-span. However based on the data points they show, it seems reasonable to question if one would expect a factor of 2 drifting at the time scale of a day. Thus at this work's critical frequency ω_E , a large uncertainty due to a not-perfect measurement equipment itself should be expected. It would be really helpful if the authors can elaborate on either how such effects are already included in the noise spectrum or why doesn't it contribute significantly to the results.

Our reply:

We thank the reviewer for their comment. We have investigated the calibration factor stability over shorter timescales and found that fluctuations within the 25 h window contribute to approximately 10% uncertainty of the calibration factor. A detailed discussion of the setup stability is provided in Chapter 6 of Ref. (Padniuk, 2024). We believe that the dominant factor-of-2 variation observed in Fig. 7 reflects long-term drifts rather than short-term random fluctuations around a mean value. Therefore, accounting for this drift adequately captures the main source of calibration uncertainty relevant to our analysis.

Padniuk, M., *Wide-frequency-range atomic comagnetometry to search for spin-dependent interactions beyond the Standard Model*, Ph.D. thesis, Jagiellonian University in Kraków, Poland (2024). It can be found here:

<https://ruj.uj.edu.pl/server/api/core/bitstreams/27ee1f34-d5be-4094-afae-17a1c53df607/content>

Action taken:

We added a sentence referring to M. Padniuk's dissertation (line 575):

A detailed discussion of the setup stability on shorter time scales is provided in Chapter 6 of Ref. [16].

Comment 3 and 4:

I want to thank the authors for explaining Fig.9 and also Eq.(17). It seems that they are both accounting for the Monte Carlo process characterizing the stochastic behavior of axion background. It is still not very clear to me though, why it is necessary to implement Monte Carlo to determine the spatial derivative and phase status of the axion background, which seems to be one of the innovation point of this paper. It can be helpful if a line representing a standard virialized, isotropically distributed, evenly phase-distributed axion background can be drawn in a relevant plot, showing the advantage of Monte Carlo method.

It seems that there would be a most conservative constraint, by assuming the axion background is at an extreme phase where the theoretically induced signal would be smallest, i.e. suppressed by $m_a T (=92 \text{ days})$. The authors seem to acknowledge such existence and take a Monte Carlo method which is already beyond the standard of this field. However it can be interesting to know where the most conservative constraint would be.

Our reply:

Equation 17 characterizes the amplitude of the ALP field when measuring for less than a full oscillation. In this situation, there would be an ALP signature at the sidereal frequency due to Earth rotation which modulates the ALP field in the interferometer. However, the amplitude during the measurement could be a fraction of the total amplitude of the ALP oscillation depending on the phase. With the factor κ this is taken into account while setting limits. We take a 95% C.L. as it is usual when searching for new particles, but if we were more conservative the limits would be weaker.

Action taken:

We have added a new figure with a caption where we plot the factor κ in function of the Compton frequency of the ALP. In line 666 we have added: **"Figure 10 represents the 95% C.L. of the factor κ depending on the ALP frequency."**

Third Referee

Comment 1:

It was mentioned that comagnetometers have already established stringent limits in certain mass ranges. I was not sure if the results of this work constitute the first constraints from an interferometer composed of two comagnetometers.

Let me clarify further. This concern relates to the paper's novelty and positioning within the field. In particular, I believe the introduction (or abstract) could more clearly articulate: the specific gap in existing knowledge this work addresses; how this approach differs from previous comagnetometer studies (e.g., compared to the K-3He systems

shown in Figs 4 and 5); the advantages this new approach offers compared to existing methods.

I believe this change would enhance both the impact of the paper and readers' understanding of its importance in the field.

Our reply:

We agree that the novelty is not so much emphasized in the introduction, and a mention should be included. We do discuss the sensitivity improvement compared to single sensor at the end of Sec. IIA. We have added a citation to our recent conference paper where we discuss the effect of multisensors in more detail.

Action taken:

We have added in the introduction (line 74): “The interference increases the signal-to-noise ratio of the search compared to a single sensor [24]. This work constitutes the first constraints from an interferometer composed of two comagnetometers.”

[24] D. Gavilan-Martin, G. Łukasiewicz, D. F. Jackson Kimball, S. Pustelny, D. Budker, and A. Wickenbrock, Notes on optimizing a multi-sensor gradient axion-like particle dark matter search, PoS COSMICWISPers2024, 041(2025).

We thank the reviewers for accepting our manuscript for publication in Nature Communications.

Sincerely,

Daniel Gavilan-Martin and Grzegorz Lukaszewicz on behalf of the authors